# Augmenting Industrial Maintenance with LLMs: A Benchmark, Analysis, and Generalization Study

## Abstract

Monitoring the life cycle of complex industrial systems often relies on expertly curated temporal conditions derived from sensor data, a process that requires significant time investment and deep domain expertise. We explore the potential of utilizing Large Language Models (LLMs) to generate context-aware and accurate recommendations for maintenance based on their ability to reason and generalize on temporal sensor conditions. To this end, we formulate a novel pipeline that systematically converts human-authored symbolic conditions into a multiple-choice question answer (MCQA) dataset. We apply our pipeline by creating DiagnosticIQ, a 6,000+ MCQA dataset covering 16 different types of physical assets that represent real-world maintenance use cases. We assess 19 state-of-the-art Large Language Models (LLMs) with this dataset and create a leaderboard for the maintenance action recommendation task. Furthermore, we evaluate and demonstrate the practical utility of DiagnosticIQ in two key aspects. First, as a knowledge base to enhance maintenance action recommendations, and secondly, as a fine-tuning resource to fine-tune a specialized LLM that generalizes across previously unseen assets to facilitate the rule creation process.

## 1 Introduction

Industrial complex equipments such as wind turbines, air handling units, and chillers require significant domain expertise for appropriate and effective operations, maintenance and tuning. These equipments are frequently deployed in operationally critical environments, such as health care organizations and large data centers where enhancing operational reliability and efficiency are critical. To achieve this, many Industrial facilities have integrated automated monitoring systems such as Internet of Things (IoT) solutions which continuously capture sensor data reflecting the operational state of the equipment and the interconnected elements of the equipments. While these systems can detect anomalies by monitoring predefined conditions, they generally provide limited guidance on appropriate corrective actions once issues are identified.

Consider Bob, a facility manager responsible for maintaining HVAC systems in a data center. Bob configures rule-based alarms by analyzing sensor data and asset metadata, defining asset-specific logical conditions such as Temperature $> 80°$F or Enthalpy $< 15$ BTU/lb

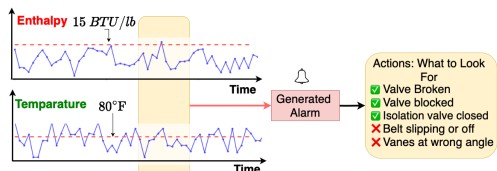

$\underbrace{\text{Temperature} > 80°\text{F}}_{\text{Condition 1}}$ or $\underbrace{\text{Enthalpy} < 15 \text{ BTU/lb}}_{\text{Condition 2}}$

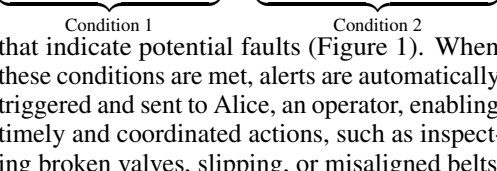

that indicate potential faults (Figure 1). When these conditions are met, alerts are automatically triggered and sent to Alice, an operator, enabling timely and coordinated actions, such as inspecting broken valves, slipping, or misaligned belts.

Figure 1: Faults to Fixes: Operation Workflow from Monitoring IoT Stream to Alarm Generation to Actionable Maintenance Recommendations

At the center of this workflow lies the configuration of recommended actions or inspection steps deciding what to look for or what maintenance should be performed once an alert is triggered. Despite timely notifications of abnormal conditions, determining the specific maintenance, repair,

or verification steps often exceeds Bob's expertise. This challenge is magnified in operational environments where there is a variety of assets from different manufacturers, with different operating modes, equipped with hundreds of sensors, resulting in a vast number of conditions to monitor simultaneously. For each abnormal condition, identifying what to inspect or repair requires specialized knowledge of asset-specific failure modes and mechanical systems expertise typically gained through years of hands-on experience. Can LLMs help bridge this gap?

Recognizing this challenge, intelligent recommendation systems that translate complex sensor data into actionable maintenance steps are critical for effective industrial asset management. Use of LLMs in discovering rules in an automated manner from labeled operational data is demonstrated recently (Zhang et al., 2025b), and motivated in recent survey articles (Raza et al., 2025; Su et al., 2024). Clearly, this stream of work will help in industrial applications such as predictive maintenance and signal monitoring (Cook et al., 2019; Kanawaday & Sane, 2017; Beghi et al., 2016; Shah & Tiwari, 2018). However, the key step of connecting these discovered rules to actionable guidance for technicians remains unaddressed. Recent advances in language models offer promise for this *action recommendations* task (Zhong et al., 2024), determining the correct maintenance or repair actions following alarms but their systematic evaluation is hindered by a lack of realistic, standardized benchmarks.

To address this gap, we present DiagnosticIQ, a novel benchmark suite for industrial maintenance action recommendation. Grounded in real-world scenarios, this suite features a primary multi-choice question-answering dataset along with several variants, each designed to systematically evaluate a specific capability vital for LLMs in this domain, such as reasoning, generalization, and robustness. We further analyze a set of frontier models under these axes to establish a strong baseline revealing the current strengths and limitations of LLMs under this domain.

**Our contributions are as follows:**

- We formalize and implement a novel deterministic dataset generation pipeline that converts expert-authored symbolic rules into MCQA dataset

- We release DiagnosticIQ and its specialized variants, a first-of-its-kind benchmark dataset with about 6,690 MCQA, expertly validated, based on 120 operational rule-action pairs.

- We benchmark 19 large language models (including Claude, Gemini, GPT variants) and establish a Maintenance Action Recommendation Leaderboard to foster community evaluation and progress.

- We systematically evaluate LLMs underaxes of Reasoning, Generalization, Robustness and demonstrate the utility of DiagnosticIQ as an external knowledge base as well as a finetuning resource for the task of maintenance action recommendation.

## 2 RELATED WORK

Building QA datasets in specialized domains has been an emerging trend across the board such as telecommunications (Lee et al., 2024), climate (Schimanski et al., 2024), finance (Chen et al., 2024), healthcare (Ray et al., 2024; Sviridova et al., 2024), IT operations (Zhang et al., 2024a), power plants (Hong et al., 2024), and scientific disciplines (Bhattacharjee et al., 2024). We review the most relevant papers with a particular focus on multi-choice QA, statistical and fine-tuning methods, rule generation and the role of domain experts.

Multi-Choice Question Answering (MCQA) has become a popular construct in the LLM world due to its ease of evaluation and closed form. This is evident from dozens of recent works such as TruthfulQA (Lin et al., 2022), GPQA (Rein et al., 2023), MMLUPro (Wang et al., 2024), FailureSensorIQ (Constantinides et al., 2025), Multi-Modal QAs (Yi et al., 2025), and Multi-Modal AD (Jiang et al., 2025). For industrial domains with maintenance tasks, we have witnessed parallel efforts such as aviation safety QA (Zhang et al., 2025a) and log classification using taxonomy (Zhang et al., 2025a; Stewart et al., 2023). PHM-Bench (Yang et al., 2025b) is another parallel effort, focusing on code-generation–based PHM tasks across 18 asset classes, whereas our work targets temporally grounded operational conditions and the maintenance actions technicians actually take. These datasets are designed to evaluate various aspects of LLM/LVMs (and in some cases embedding models), such as common-sense reasoning, domain understanding, and multimodal context. The key

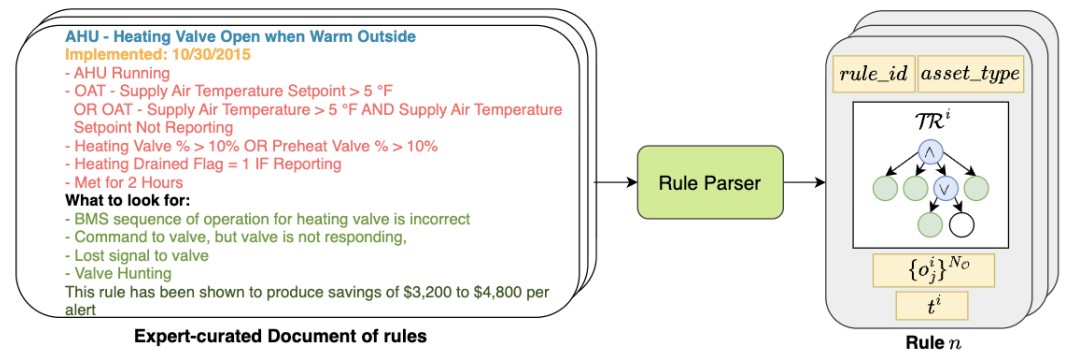

Figure 2: Example expert-curated rules and associated conditions/metadata for dataset construction.

difference lies in how these questions are generated in the first place (using existing documents or completely written by experts) and then validated by a domain expert if needed. We observed that the community has paid less attention to cases where **part of the question is presented as a rule**. ComplexBench (Wen et al., 2024) is closest to our work in term of instruction, as it highlights the importance of evaluating LLMs on their ability to follow complex instructions such as And, Selection, and Chain. However, it does not explicitly cover rules. As discussed in (Zhang et al., 2025b), rules encode domain knowledge effectively and benefit LLMs, and interestingly, many industrial monitoring systems utilize human-crafted rules for monitoring tasks. Therefore, we focus on building an MCQA dataset where the question part contains a rule.

Given an answer as part of the choices in MCQA, an LLM model may make a random selection and still achieve a favorable result. This limitation has motivated the research community t o develop innovative statistical solutions and tools, such as validation tests (Zheng et al., 2024; Zhang et al., 2024c; Robinson et al., 2023), PertEval (Li et al., 2024a; Ye et al., 2024), and other approaches (Zhu et al., 2024). The development of new, unseen MCQA datasets has further benefited the community. Furthermore, MMLUPro has adopted a method of increasing the number of options from four to ten and demonstrated that this adjustment has a non-trivial impact. However, constructing a robust solution for such complex questions remains an open challenge, necessitating innovations such as the adoption of a recommendation-based module, as discussed in this paper.

Statistical analysis of MCQA datasets still requires an additional level of human or domain expert validation, particularly for mission-critical applications. In most domain-specific QA datasets, experts are employed to participate in the quiz to quantify the difficulty of the prepared questions. Very limited analysis has been conducted on truly exercising the validation of the LLM's reasoning abilities, such as in the medical domain (Sviridova et al., 2024). Evaluating the generated rationales/reasoning not only ensures correct answers but also builds confidence in the model's outputs. Unlike most prior studies, we examine whether LLMs generate accurate rationales alongside correct answers, evaluating explanations against end-user needs rather than relying solely on accuracy.

Operational rules are common in industrial domains, as demonstrated by Oracle's Maintenance Cloud Service (Oracle, 2025), but they require significant expert involvement (Zhang et al., 2025b). For many enterprise customers, smaller language models will be key, as they provide a practical way to embed domain-specific knowledge directly into the model. Methods such as Supervised Fine-Tuning (SFT) and Group Relative Policy Optimization (GRPO) (Shao et al., 2024) have become key methods of this process, making generalization tests essential. Industrial settings often demand transferring rules written for one physical asset to another, further complicating the task. Despite the importance of these challenges, systematic comparisons in Industry 4.0 contexts remain limited, motivating our cross-asset knowledge transfer evaluation.

## 3 SYMBOLIC CONDITIONS TO MCQA

This section details our methodology, beginning with the introduction of our pipeline that systematically converts symbolic, human-authored rules into a MCQA format. We then describe its application in the domain of industrial asset maintenance to create our benchmark dataset, DiagnosticIQ. The generation pipeline consists of three primary stages: (1) parsing rule documents into structured representations, (2) converting these representations into Disjunctive Normal Form (DNF), and (3) Selecting sets of actions. Finally, we discuss the development of several variants of DiagnosticIQ, each tailored to evaluate a specific capability of LLMs such as their temporal reasoning and generalization that is critical for the maintenance recommendation task.

### 3.1 INPUT: EXPERT-CURATED RULE DOCUMENTS

The rule documents originated from the Smarter Buildings initiative, where Reliability Engineers, System Administrators, and a Rules Logic committee collaboratively developed and iteratively refined fault-detection logic across multiple equipment types (e.g., Air Handlers, Chillers, Boilers) over several years to expand coverage and maintain diagnostic accuracy. A detailed description of the rule development process is provided in Appendix E which is motivated from guideline (ASHRAE, n.d.). The pipeline generation process begins by extracting these expert-defined rules from the active monitoring system (Figure 2), where domain experts author the conditions that trigger maintenance actions. Each rule $\mathcal{R}^i$ typically comprises three key components: (1) a set of **conditions** that must be satisfied for a specified duration ($t^i$) to activate the rule. Let $\mathcal{C}^i = \langle c_1^i, c_2^i, \ldots, c_n^i \rangle$ be the set of atomic boolean conditions associated with it, where each $c_j^i$ is a predicate over sensor readings or asset states (e.g., Temperature $> 80°$F, Enthalpy $< 15$ BTU/lb); (2) a set of **maintenance actions** for a rule ($\mathcal{O}^i = \{o_1^i, o_2^i, \ldots, o_{N_O^i}^i\}$ where $N_O^i$ represents the total number of actions), hypothesized for verification once triggered; and (3) **metadata** including rule id (*rule_id*), rule name, asset type *asset_type* (e.g., Air Compressor, Boiler), rule description, and estimated cost savings (in dollars) achieved by applying rule ($\mathcal{C}^i$).

From the domain documentation, we extract and assemble these conditions ($\mathcal{C}^i$) into a structured logical formula, referred to as a *condition tree* $\mathcal{TR}^i$. This tree is a boolean expression constructed from $\mathcal{C}^i$ using logical operators $\wedge$ (AND), $\vee$ (OR), and optionally $\neg$ (NOT). Formally: i) Each leaf node in $\mathcal{TR}^i$ corresponds to an atomic condition $c_j^i$, ii) Internal nodes represent logical operators from the set $\{\wedge, \vee, \neg\}$, iii) The root node evaluates to True if the entire condition tree is satisfied given the current sensor state. Given the above formulation, we define a rule $\mathcal{R}^i$ as the tuple:

$$\mathcal{R}^i = (rule\_id,\ asset\_type,\ \mathcal{TR}^i,\ \mathcal{O}^i,\ \mathcal{C}^i,\ t^i)$$

We denote the collection of expert-written rules denoted as $\mathcal{DS}_{\mathcal{R}} = \{\mathcal{R}^i\}_{i=1}^{N_{\mathcal{R}}}$, where $N_{\mathcal{R}} = 120$. The rules span several asset types (10+) listed in Table 6. Table 6 summarizes, for each asset type, the number of rules (#Rules), the number of disjunction operators (#$\vee$), the total number of atomic conditions (#$\mathcal{C}$), and the number of observations. The count of disjunctions (#$\vee$) is particularly informative, as it reflects the branching complexity within the condition trees $\mathcal{TR}^i$, enabling us to sample a diverse range of conditions to which each rule $\mathcal{R}^i$ applies.

To provide additional context on these industrial assets, we developed concise descriptions (Desc) for each asset type in collaboration with industry experts (Appendix Table 10). These descriptions are incorporated into the question-generation process to improve the relevance and clarity of the dataset.

### 3.2 QA GENERATION PIPELINE

We design two primary types of diagnostic questions to evaluate a language model's reasoning capabilities under varying constraints. The first type requires the model to identify the **most relevant** option given the Question Conditions $QC$, testing its ability to prioritize the most probable root cause based on domain-specific knowledge. The second type requires selecting the **least relevant** option, challenging the model to distinguish between superficially similar answers and recognize when a condition-action mapping is unsupported by the available evidence. We refer to these question types as **selection** and **elimination**, respectively. Prior to question generation, we compute the following metadata to support dataset construction:

**i) Rule-Rule Similarity (RRSim)**: For each rule $\mathcal{R}^i$, we construct a textual representation of its condition tree $\mathcal{TR}^i$ based on expert-authored documentation. We then generate embeddings for these texts and calculate pairwise cosine similarity scores across all rules. This metric enables sampling of semantically similar or diverse rules when constructing questions.

**ii) Unique Observations (UO)**: We manually curate and categorize observations/actions across the rule set $\mathcal{DS}_\mathcal{R}$ to identify a universal set of unique observations. These form a candidate pool for selecting answer options. At present, $|UO|$ = 193 and average length of each observation is around 20 (See Appendix 9 for distribution).

### 3.2.1 QUESTION-ANSWER STRUCTURE

Each question $\mathcal{Q}^i$ in the dataset is represented as a tuple $(AD, QC, QP, OPT, A)$, where $AD$ denotes the asset name along with its description obtained from Desc, $QC$ specifies the observed conditions exhibited by the asset in the context of the question, $QP$ represents the question prompt; further details on its construction are provided below, $OPT$ is the set of answer options around 4 or 10. and $A$ indicates the ground-truth correct answer for the question (Single true in at present).

### 3.2.2 RULE REPRESENTATION TO DISJUNCTIVE NORMAL FORM (DNF)

The QA generation procedure is summarized in Algorithm 1 (Appendix N). For each rule $\mathcal{R}^i$, we first convert the condition tree $\mathcal{TR}^i$ into its *Disjunctive Normal Form (DNF)* that is, a disjunction (OR) over conjunctions (ANDs) of atomic conditions:

$$\mathcal{TR}^i_{\text{DNF}} = \bigvee_{k=1}^{K} \left( \bigwedge_{j=1}^{m_k} c^i_{kj} \right)$$

Each conjunctive clause in this DNF represents a *complete and specific observation pattern* sufficient to activate the rule $\mathcal{R}^i$. This formulation allows us to consider $K$ distinct conjunctions that satisfy $\mathcal{TR}^i_{\text{DNF}}$ = True.

For example, consider a rule $\mathcal{R}^0$ with atomic conditions: $c_1$ = (Preheat Valve% $\geq$ 5%), $c_2$ = (Heating Valve% $\geq$ 5%) and $c_3$ = (Heating Drained Flag = 1 if reporting) let the condition tree be $\mathcal{TR}^0 = (c_1 \vee c_2) \wedge c_3$ and the and its DNF form $(c_1 \wedge c_3) \vee (c_2 \wedge c_3)$. Each conjunction, $(c_1 \wedge c_3)$ and $(c_2 \wedge c_3)$, is treated as a distinct, fully instantiated observation scenario, which is used as a unique $QC$ for generating a question. This systematic transformation ensures that each QA instance is grounded in logically valid asset state combinations, reflecting domain expert reasoning and enabling scalable, interpretable dataset construction.

### 3.2.3 SELECTING SETS OF ACTIONS

Next, we select observation combinations to construct the answer options for both *selection* and *elimination* question types. For *selection*-type questions (extracted_obs_sel in Algorithm 1), we identify candidate incorrect options by retrieving the $N_{sel\_topk}$ least similar rules to $\mathcal{R}^i$ using **RRSim** and collecting their observations, denoted as $INC^i = \{inc^i_j\}_{j=1}^{N_{inc}}$. We then generate answer tuples $\{(QP^i_j, OPT^i_j, A^i_j)\}_{j=1}^{N_{sel}}$ by:

(1) Selecting each $o^i_j \in \mathcal{O}^i$ as the correct option, (2) Random sample $\alpha$ incorrect options $\in INC^i \backslash \mathcal{O}^i$,

(3) Composing the prompt $QP^i_j$, which is drawn randomly from a pool of $N_{QT}$ question templates.

For *elimination*-type questions (extracted_obs_eli in Algorithm 1), the correct options correspond to observations that do not belong to $\mathcal{R}^i$. Specifically, we retrieve the $N_{ele\_topk}$ least similar rules to $\mathcal{R}^i$ using **RRSim** and collect their observations, denoted as $COR^i = \{cor^i_j\}_{j=1}^{N_{cor}}$. We then randomly sample $\min(N_{cor}, \beta)$ observations from $COR^i$ as correct options. For each correct option, we construct a question by pairing it with incorrect options sampled from $\mathcal{O}^i$, ensuring the elimination task challenges the model to identify irrelevant actions in the context of the given conditions.

$\alpha$ and $\beta$ are hyperparameters controlling the number of questions per rule. Larger values increase question count but reduce diversity, while smaller values enhance uniqueness but limit coverage.

# 4 DIAGNOSTICIQ

We apply the pipeline described in 3 on 120 expert curated rules to create DiagnosticIQ. we set the hyperparameters $N_{sel\_topk} = 25$, $N_{eli\_topk} = 25$, $N_{QT} = 10$, $\alpha = 10$ and $\beta = 10$ during the creation process. The final dataset contains 6690 questions, with asset composition detailed in Figure 3. Selection-based questions make up 77.4% of the dataset, compared to 22.6% elimination questions. As shown in Figure 3 the majority of QA instances focus on AHU related scenarios (58.2%), followed by Chiller and Fan (5.9%)

The predominance of selection-type questions arises from the limited sample space for generating incorrect options in elimination questions, which rely on the set $\mathcal{O}^i$ for a given $\mathcal{R}^i$. Although this imbalance could be adjusted by setting $\alpha \ll \beta$, as previously discussed, doing so risks generating many similar elimination questions, ultimately reducing the diversity of the dataset.

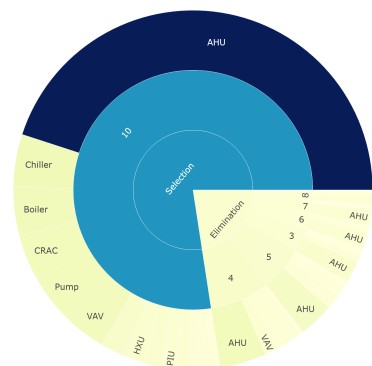

Figure 3: DiagnosticIQPro Composition by Asset/Number of Options/Question Type

The dataset intentionally reflects a real-world class imbalance, with a majority of rules pertaining to Air Handling Units (AHUs), as shown in Figure 3. We preserve this skew rather than resampling to accurately model operational priorities. Furthermore, while 'selection' questions have a fixed size of 10 options, 'elimination' questions feature a variable number of choices. This is because the incorrect options for elimination questions are drawn from the smaller, contextually relevant pool of alternatives available within the original source rule.

```json
{
  "AD": "Closed-loop water-cooled chiller system with cooling tower.",
  "QC": ["Chiller Running","Evaporator Delta T < 7degF",
         "Cooling Tower Supply Temp < Setpoint - 4degF","OAT > 43degF"],
  "QP": "Given the observed conditions, what is the most likely root cause?",
  "OPT": ["(A) Cooling tower is overcooling condenser water",
          "(B) Chiller evaporator is fouled",
          "(C) Supply water pump is cavitating",
          "(D) Building load is too low"],
  "A": "(A)"
}
```

Listing 1: DiagnosisIQ QA Instance for Chiller–Tower Case

## 4.1 VARIANTS

We tweak DiagnosticIQdataset to create several variants that test LLMs under different conditions.

**(1) DiagnosticIQPro** : We features a wider range of answer choices, with 10-option (increasing $\alpha$ and $\beta$ up to 10) questions constituting the majority (77.4%), thereby increasing dataset complexity (Figure 3). This distribution aligns with both practical asset relevance and intentional design decisions to balance diagnostic depth with question difficulty.

**(2) DiagnosticIQPert** : We Perturb MCQ in DiagnosticIQ utilizing PertEval benchmark (Li et al., 2024a) (Appendix N.3). We use this dataset to evaluate robustness against formatting.

**(3) DiagnosticIQRationale** : We develop a dataset that has the rationale the LLM follow to arrive at the answer for MCQA questions, this is used to conduct human evaluation of the expert knowledge an LLM posses in the domain of maintenance recommendation. Further we utilize this for finetuning for the generalizability study.

**(4) DiagnosticIQVerbose** : To identify the effect of presenting $QC$ in natural language, we develop a variant of DiagnosticIQ that converts the symbolic representation of $QC$ to natural language. The procedure for conversion can be found in (Appendix N.4)

Table 1: Leaderboard: DiagnosisIQ and Pro. (* indicates closed source models)

| Rank | Model | Macro. Diag IQ | Macro. +Pro | Diag IQ | +Pro |
|------|-------|----------------|-------------|---------|------|
| 1 | claude-3-7-sonnet* | **70.61** | **56.63** | **72.66** | **53.80** |
| 2 | deepseek-v3 | 67.02 | 41.38 | 67.89 | 35.80 |
| 3 | o1* | 65.41 | 24.79 | 70.22 | 26.11 |
| 4 | mistral-large | 63.15 | 41.13 | 65.52 | 36.50 |
| 5 | qwen2-5-72b-ins. | 61.22 | 35.91 | 63.09 | 32.93 |
| 6 | llama-3-3-70b-ins. | 61.67 | 36.56 | 60.33 | 32.27 |
| 7 | mistral-small-3-1-24b | 61.17 | 33.79 | 60.15 | 28.42 |
| 8 | mistral-medium-2505 | 60.34 | 35.36 | 61.43 | 30.16 |
| 9 | granite-3-3-8b-instruct | 59.45 | 42.39 | 57.26 | 31.43 |
| 10 | gemini-2.0-flash* | 57.64 | 26.65 | 54.63 | 20.82 |
| 11 | llama-3-1-405b-ins. | 56.56 | 38.82 | 59.03 | 35.58 |
| 12 | gemini-1.5-pro* | 53.14 | 24.72 | 65.44 | 27.77 |
| 13 | microsoft-phi-4 | 50.52 | 31.35 | 47.50 | 23.99 |
| 14 | claude-3-5-haiku* | 46.93 | 17.72 | 44.41 | 15.55 |
| 15 | llama-3-1-8b-ins. | 38.69 | 18.80 | 36.70 | 12.89 |
| 16 | claude-4-sonnet* | 62.52 | 33.44 | 68.15 | 32.99 |
| 17 | gemini-2.5-pro* | 57.59 | 37.51 | 63.44 | 38.85 |
| 18 | gpt-5-2025-08-07* | 65.89 | 40.69 | 67.79 | 40.39 |
| 19 | qwen3-8b | 46.21 | 19.70 | 43.41 | 14.65 |

## 5 EXPERIMENTAL RESULTS

We perform a direct zero-shot prompting of the generated questions to assess the reasoning capacity of the LLMs (representative examples are provided in Appendix Figure 8). Our evaluation identifies areas of underperformance under zero-shot conditions. For evaluation we consider the Accuracy and the Macro-Accuracy as the main evaluation metrics. We utilize the Macro accuracy as DiagnosticIQ has a Asset class imbalance and will be used. Accuracy as $\text{Acc} = \frac{1}{|D_{\mathcal{Q}}|} \sum_{x \in D_{\mathcal{Q}}} \mathbb{K}\left[M(q(x)) = y_x\right]$ and Macro accuracy as $\text{Acc}_{\text{macro}} = \frac{1}{|A|} \sum_{a \in A} \frac{1}{|D_a|} \sum_{x \in D_a} \mathbb{K}\left[M(q(x)) = y_x\right]$ where $q(x)$ is the generated prompt, $M(q(x))$ the model's response, $\mathbb{K}[\cdot]$ the indicator function returning 1 for correct predictions and 0 otherwise, $D_{\mathcal{Q}}$ represents the whole dataset, $A$ is the set of assets and $D_a$ represents the questions belonging to asset class $a$.

### 5.1 LEARDERBOARD

The comparative results in Table 1 show Claude-3-7-Sonnet achieving the highest Macro accuracy on both tasks (70.61% on DiagnosisIQ and 56.63% on DiagnosisIQPro), with larger models like Mistral-Large also performing well, while smaller models such as LlaMA-3-1-8B lag behind, indicating a clear correlation between model size and performance. These findings establish a strong baseline, revealing that general-purpose LLMs struggle with reasoning about sensor conditions in industrial settings, and the sharp drop in accuracy on DiagnosisIQPro highlights the challenge of larger, realistic action spaces. Overall, the leaderboard demonstrates a pronounced performance gap favoring larger models and underscores the critical need for domain-specific knowledge integration to enable effective real-world industrial diagnostics.

Apart for Claude-3-7-Sonnet, the performance of all remaining 10 models on the DiagnosisIQPro dataset is below 45%, revealing a surprising gap on this complex task and underscoring the universal need for improvement in handling industrial diagnostic reasoning. A detailed analysis in Appendix Tables 15 and 16 shows that, across model families, incorrect predictions consistently exhibit larger set-size deviations than correct ones, indicating systematic differences in error severity.

**Embedding-based Baselines.** To establish non-generative baselines, we evaluate several widely used sentence-embedding models. Each MCQA item consists of a question stem (text concatenation of $AD$, $QC$, $QP$) and a set of candidate options. For every question, we compute embeddings for both the question and each answer choice using a given model, and select the predicted answer by maximizing cosine similarity between the question embedding and each option embedding. This

retrieval-based formulation provides a simple and assumption-free baseline that does not rely on task-specific training.

Table 2 summarizes performance across four embedding models. The best-performing model, `all-mpnet-base-v2`, reaches a $\text{Acc}_{\text{macro}}$ of 52.73% and accuracy of 41.39%. Although lightweight models such as `all-MiniLM-L6-v2` and `all-distilroberta-v1` achieve similar results, the overall accuracy remains only modestly above chance for multi-option MCQA. These same embedding models were used in our dataset construction pipeline to select distractor options. Their limited performance therefore reinforces a key design insight: semantic similarity alone is not sufficient for reliably solving our MCQA tasks. This further highlights the difficulty of the benchmark and motivates the need for more advanced reasoning-capable LLMs and agentic approaches.

Table 2: Embedding-based baseline performance on MCQA tasks.

| ID | Model | Macro. DiagIQ | Macro +Pro | DiagIQ | +Pro |
|----|-------|---------------|------------|--------|------|
| 1 | `all-mpnet-base-v2` | 52.73 | 38.89 | 41.39 | 23.39 |
| 2 | `all-MiniLM-L6-v2` | 52.65 | 37.76 | 41.32 | 23.01 |
| 3 | `multi-qa-mpnet-base-dot-v1` | 51.43 | 37.53 | 38.93 | 21.54 |
| 4 | `all-distilroberta-v1` | 51.29 | 36.98 | 38.53 | 23.47 |

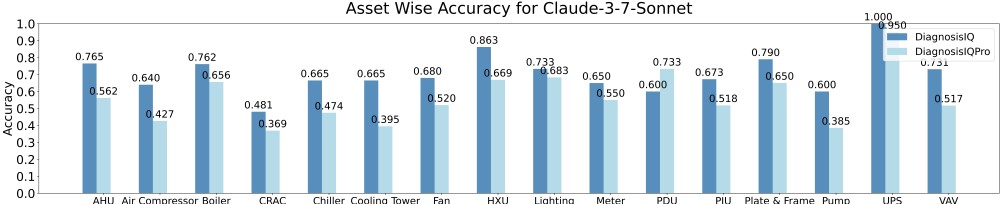

Figure 4: The Accuracy variation across varies industrial assets for claude-3-7-sonnet

**Asset-wise Analysis** We select Claude-3-7-sonnet from the leaderboard for asset-wise analysis. Figure 4 shows asset-wise accuracy comparisons, revealing consistently higher accuracy on assets like UPS 100.0%) and HXU 86.3%) in DiagnosisIQ, but analysing DiagnosisIQPro considerably drops overall, where HXU drops by -19.4% (to 66.9%). Further for CRAC (-11.2%), Cooling Tower (-27.0%), and Pump (-21.5%). These results highlight the model's domain sensitivity, with some assets maintaining robust performance, and expose a reasoning complexity gap, where accuracy declines sharply in more challenging, multi-condition scenarios.

**Question type wise Analysis**. Figure 11 compares model accuracy on selection and elimination tasks across both datasets. Elimination consistently yields higher accuracy (e.g., Mistral-Large achieves 71.0% on elimination-DiagnosisIQ vs. 63.9% on selection-DiagnosisIQ). However, the performance drop from DiagnosisIQ to DiagnosisIQPro is more pronounced for selection questions (Mistral-Large drops from 63.9% to 28.9%) than for elimination questions (71.0% to 62.6%), indicating that compositional reasoning in complex scenarios is particularly challenging when models must select the most relevant option. The same patterns seems to be consistently shown in multiple models.

**Robustness Against Perturbation.** MCQA datasets inherently contain bias due to planted correct answers. To assess this, we applied perturbation analysis on DiagnosisIQ using the PertEval benchmark (Li et al., 2024a) (Appendix N.3), generating perturbed questions and measuring accuracy (Acc) on both original and perturbed sets. Consistency accuracy was computed as Acc@Consist $= \frac{1}{|D_Q|} \sum_x C(x) \wedge C_{\text{perturb}}(x)$, where $C_{\text{perturb}}(x)$ indicates correct predictions on perturbed prompts. Table 3 reports Acc@Perturb, perturbation drop rate (PDR), and Acc@Consist, highlighting variation in model robustness to question perturbations.

Table 3: PertEval, Significant ** ($\alpha = 0.01$)

| Model | Acc@Per. | PDR | Acc@Con. |
|-------|----------|-----|----------|
| llama-3-3-70b | 66.77 | 0.11** | 52.30 |
| deepseek-v3 | 66.70 | -0.02** | 57.66 |
| qwen2-5-72b | 62.61 | -0.01 | 52.28 |
| llama-3-1-405b | 60.66 | 0.03** | 48.09 |
| mistral-medium | 60.00 | -0.02** | 49.45 |
| mistral-large | 57.39 | -0.12** | 49.92 |
| micro.-phi-4 | 45.36 | -0.04** | 32.00 |
| llama-3-1-8b | 44.03 | 0.20** | 23.01 |

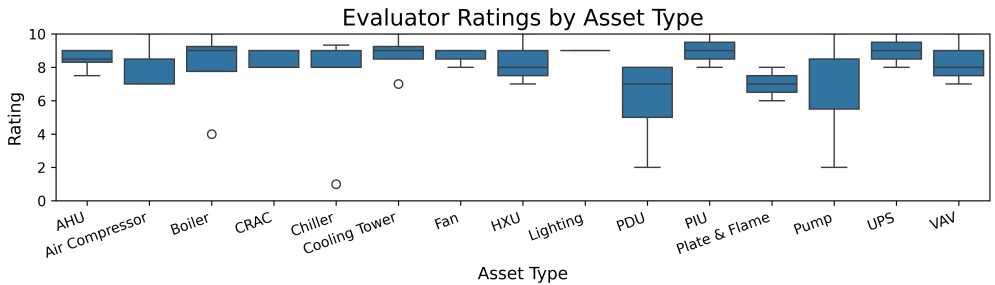

Figure 5: The expert rating for reasoning under mistral-large

## 5.2 EVALUATING DOMAIN UNDERSTANDING AND REASONING

To assess the domain understanding and reasoning patterns of LLMs, we conducted a human evaluation of the rationales they generate for maintenance recommendations. We prompted Mistral-Large to provide its reasoning given a question and correct answer on a representative sample of 27 questions from DiagnosticIQ (example rationale Fig. 12 and 13), ensuring at least one question from each asset type. We explicitly provide the correct answer as we are specifically evaluating whether the reasoning patterns of an LLM can match a domain expert. We then had five domain experts rate each generated rationale on a scale of 0 (incorrect) to 10 (expert-level quality).

As shown in Figure 5, the analysis reveals that the model's reasoning generally aligns with expert reasoning, supporting its potential to augment maintenance tasks. However, for certain assets (e.g., PDU, Pump, Boiler), we observe inter-rater disagreement, with expert feedback indicating differing expectations on the required granularity of the explanation. Furthermore, the model consistently scored lower among evaluators on the Plate & Frame HX asset, with experts expressing that the rationales lack the nuanced operational knowledge required, signifying a potential knowledge gap for that specific equipment type. These concerns must be addressed when deploying LLMs in this task.

## 5.3 GENERALIZABILITY STUDY

Transfer learning between different assets shows promising results industrial automation (Maschler & Weyrich, 2021) for tasks such as fault prediction or anomaly detection. However, transfer learning of rules between different industrial assets is an interesting direction that has not been studied. We consider Qwen3-8B (Yang et al., 2025a), Llama-3.1-8B-Instruct (Dubey et al., 2024), and Granite-3.3-8B-Instruct (Granite Team, 2024). To avoid any data leakage, the split is stratified by asset. To account for the imbalance in the questions per asset (Figure 3), we consider two splits: AHU and the rest of the assets. For each model we finetune on each split and test on the other split. We use Supervised Fine-Tuning (SFT) and GRPO. Overall, the rule learning is transferable across assets, with Qwen3-8B being the best performing model both with and without SFT. More details on model specific prompt formatting in Appendix Section I. As shown in Table 4, both SFT and GRPO fine-tuning consistently improve micro and macro accuracy compared to base models across the AHU/Other splits. While SFT shows strong gains for Qwen3, GRPO tends to perform better on Llama3.1 and Granite3.3, indicating model- and data-specific benefits.

## 5.4 CONDITION FORMATTING STUDY

We investigate effect converting the temporal conditions into natural language as a pre-processing step as in (Zhang et al., 2025b). We utilize the Macro accuracies of the DiagnosticIQVerbose dataset (dataset creation details can be found in Appendix N.4) and the difference in Macro accuracy of DiagnosticIQ and DiagnosticIQVerbose ($\Delta$ Macro Accuracy) for this and compare the effectiveness of the formatting. We present our findings in 8. We identify that the Macro accuracies drop on almost all models which shows the symbolic understanding the LLMs possess atleast in the domain of physical asset maintenance.

Table 4: SFT/GRPO experiments on training/testing AHU/Other splits. Micro accuracy is reported in AHU/Other, along with the overall Macro accuracy.

| Model (8B) | Setting | AHU | Other | Macro |
|---|---|---|---|---|
| Llama3.1 | Base | 50.88 | 44.52 | 48.61 |
|  | SFT | 52.31 | 56.44 | 54.09 |
|  | GRPO | 52.95 | 61.76 | 61.45 |
| Qwen3 | Base | 56.47 | 66.85 | 61.63 |
|  | SFT | 68.89 | 80.28 | 72.12 |
|  | GRPO | 55.27 | 64.76 | 64.49 |
| Granite3.3 | Base | 59.02 | 56.58 | 59.99 |
|  | SFT | 58.79 | 59.61 | 59.40 |
|  | GRPO | 54.94 | 63.76 | 63.56 |

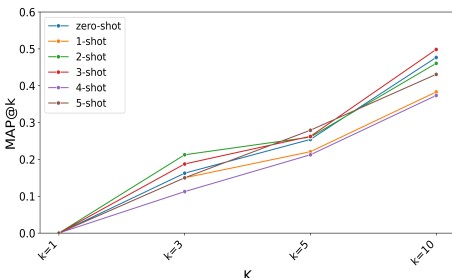

Table 5: MAReE MAP@K with number of examples from DiagnosticIQ

## 5.5 Maintenance Action Rec. Engine

We present **M**aintainance **A**ction **Re**commendation **E**ngine (MAReE), a deployed application leveraging DiagnosticIQ to recommend maintenance actions based on abnormal conditions defined by subject matter experts (SMEs). MAReE employs **LLM-Score**, which assigns relevance scores to candidate actions. We evaluate MAReE on 11 new SME-authored rules with ground truth actions, varying $k \in \{1, 3, 5, 10\}$ and measuring MAP@K (Mean Average Precision at k), indicating whether the ground truth action appears in the top-$k$ recommendations. For each rule, 10 candidate actions are dynamically sampled using an embedding model to create realistic and challenging evaluation scenarios. Results in Figure 5 show that 3-shot prompting achieves the highest MAP@10 score (49.84%), outperforming zero-shot and other shot counts, while MAP@K scores for $K > 1$ fluctuate without a clear trend, suggesting that increasing example count does not consistently improve recall beyond the top prediction, though few-shots beat the zero-shot at any k. Detailed system prompt is available in Appendix 10.

## 6 Conclusion and Limitations

This paper addresses the gap of systematic evaluation of LLMs on maintenance action recommendation in the industrial setting. We develop a generic pipeline that systematically converts symbolic, human-authored rules into an MCQA format. Utilizing our pipeline we introduce DiagnosticIQ and its variants designed to benchmark the ability of utilizing LLMs to recommend maintenance actions. We benchmark 15 leading LLMs, establishing the first standardized leaderboard for this task. Our analysis systematically evaluated model reasoning, generalization, and robustness, providing a clear baseline for the community.

Our work confirms the potential of LLMs as a powerful tool in this domain and provide resources for further developing reliable industrial AI. However, we acknowledge a few key limitations and future directions of research. The industrial domain includes over 800 asset types, but our current study covers only a limited subset again due to the extensive amount of time being spent (nearly 120 days using 3-4 SMEs) on writing each pair of condition-rule and action. We plan to expand assets coverage to improve robustness and mitigate potential biases due to LLM familiarity with specific assets. Furthermore, although out dataset generation pipeline is generic and applicable to other domains such as business process management and cloud resource monitoring our experiments so far focus on Industry 4.0. Extending this approach to additional domains remains an important future direction. We envision this work as foundational step towards more sophisticated assistive tools that augment rule creation as a whole and automated asset monitoring. The MAReE experiments underscore the need for more advance methods or finetuned models to reliably address this task.

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

## A  DECLARATION OF GENERATIVE AI AND AI-ASSISTED TECHNOLOGIES IN THE WRITING PROCESS

During the preparation of this work, the authors used Grammarly in order to improve the grammar, clarity, and flow of the manuscript. After using this tool/service, the author(s) reviewed and edited the content as needed and take full responsibility for the content of the publication.

## B  ETHICS STATEMENT

In this paper, we strictly obey the principles outlined in the ICLR Code of Ethics, including careful consideration of potential ethical concerns, including the impact on human subjects, data privacy, and fairness in dataset construction decisions. We promise that any data used in this study were released in compliance with legal and ethical standards, and proper security measures were implemented to safeguard personal and location information. The dataset hosting platform will be huggingface and/or kaggle benchmark.

## C  REPRODUCIBILITY STATEMENT

We provide the all the details of our method in the paper and appendix, including evaluation prompts, detailed experimental setup and implementation, hyperparameters for both LLM reasoning and MCQA questions. The code will be available upon the paper publication. These above ensure that others can reproduce our method.

## D  HYPERPARAMETER ANALYSIS

we analyse the effect of varying the $\alpha$ and $\beta$ to see the dataset option selection diversity as setting them relatively high may result in a larger number of questions with the tradeoff of having similar question options among questions. To quantify the diversity of the options we calculate the question to question overlap of generated options for question from several rules (rules with ids $1, 4, 9, 16, 25$) to calculate the overlap we use the Intersection over union measure (IOU) and calculate the mean IOU of the selected questions per dataset. The results are presented in the Fig. 6. which clearly shows that when we increase $\alpha$ and $\beta$ the dataset size increases (propotional to the size of the bubble) although the mean IOU value increases as well thus balancing out both the diversity and question count we choose $\alpha = 10$ and $\beta = 10$ in DiagnosticIQ

$$IOU_{mean} = \frac{1}{|D_Q|} \sum_{x_1, x_2 \in D_Q} \frac{OPT_{x_1} \cap OPT_{x_2}}{OPT_{x_1} \cup OPT_{x_2}}$$

where $D_Q$ here is a subset of the questions for the given rules for a dataset

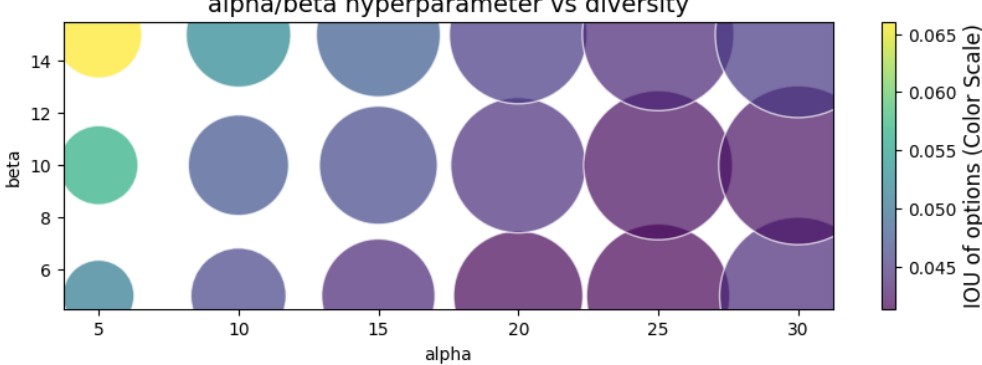

Figure 6: The Variation of Dataset size and Mean IOU as $\alpha$ and $\beta$ are Varied

# E    RULE GENERATION

The rules originally came from Smarter Buildings, which was introduced in 2011 as part of the larger Smarter Planet initiative. There were originally 18 Fault Detection and Diagnostic (FDD) rules spread across 3 equipment types, with Air Handlers being the primary focus. There were two main objectives to the program: to achieve 5-15% energy savings for the monitored equipment, and to reduce maintenance hours by 30%. Air Handlers were originally, and continue to be, the primary focus of the rules, as they are the most prevalent piece of equipment at any location and thus provide the greatest savings. As the program continued and the rule set expanded, additional equipment (such as Chillers and Boilers) was added for increased monitoring and savings, leading to a total of 118 active rules across 13 equipment types.

The rules were actively developed over the course of 7 years, with 11 major updates. Each update added additional rules and updated existing ones to account for updates to the logic. There were two key roles involved in developing the rules: the Reliability Engineer who developed the rules, and the System Administrator who coded them. There was also a Rules Logic committee that typically involved 2-8 participants that met every other week to brainstorm and develop the rough logic for the rules, with the Reliability Engineer and System Administrator working closely together to ensure the logic was coded correctly.

Writing a new rule does not take a long time to code – typically around 30 minutes to ensure it's running correctly. However, with additional testing and verification, it can take significantly longer to finalize. This includes documentation of the new rule, as well as the correct verbiage on the rule, to better inform technicians about the potential causes and where to begin troubleshooting. Updating a rule takes less time but still requires additional testing and updating of the documentation, so it is still not an insignificant task.

# F    LLMS IN INDUSTRY 4.0

Large language models (LLMs) have seen rapid development and broad application across domains, from general-purpose models like OpenAI's GPT series (OpenAI, 2024) and Meta's Llama 2 (Touvron et al., 2023) to specialized, multimodal models such as Gemini (Team et al., 2024) and Mistral 7B (Jiang et al., 2023). These foundational models demonstrate impressive capabilities in natural language understanding, generation, and reasoning (Wei et al., 2022; Wang et al., 2023; Yao et al., 2022), and are increasingly benchmarked on complex question answering and reasoning datasets (Rein et al., 2023; Li et al., 2024b; Wang et al., 2024).

In the context of domain-specific applications, several efforts highlight the benefits of fine-tuning or training LLMs on domain-relevant corpora. For example, INDUS (Bhattacharjee et al., 2024) and TelBench (Lee et al., 2024) demonstrate improved performance on scientific and telecommunications tasks, respectively, underscoring the value of specialized data and benchmarks. Similarly, clinical text models (Li et al., 2024c) and biomedical retrievers (Xu et al., 2024) leverage domain-specific adaptations to better meet task requirements.

Industrial and predictive maintenance applications have attracted increasing attention with approaches combining LLMs and machine learning for failure mode classification and condition monitoring (Stewart et al., 2023; Nikitin & Kaski, 2022; Putchala et al., 2022; Yang et al., 2022). These studies highlight the potential of language models to extract actionable insights from maintenance logs, sensor data, and domain knowledge, aiding decision-making in complex industrial environments. Public datasets such as predictive maintenance for air compressors (Okudan, 2023) and wind power forecasting (Bhaskarpandit, 2020) facilitate research in this area.

Recent work also explores the robustness and reliability of LLMs in handling structured data, multi-hop reasoning, and multiple-choice question answering (MCQA). Studies have pointed out challenges in LLMs' MCQA performance (Robinson et al., 2023; Zhang et al., 2024c; Zheng et al., 2024) and proposed methods like chain-of-thought prompting (Wei et al., 2022) and plan-and-solve prompting (Wang et al., 2023) to elicit better reasoning. Self-improving multi-step reasoning agents (Aksitov et al., 2023) and trustful frameworks for unified question answering (Zhang et al., 2024b) further advance LLM capabilities.

Table 6: Statistics of expert-curated rules collected (Asset type definitions can be found in Table 10).

| Asset Type | #Rules | #∨ | #$\mathcal{C}$ | #Observations |
|---|---|---|---|---|
| Fan | 1 | 0 | 4 | 4 |
| UPS | 1 | 1 | 2 | 1 |
| Lighting | 1 | 1 | 2 | 2 |
| Plate & Frame | 1 | 1 | 4 | 4 |
| PIU | 2 | 1 | 5 | 5 |
| Meter | 3 | 4 | 4 | 4 |
| Air Compressor | 3 | 1 | 6 | 8 |
| PDU | 3 | 0 | 7 | 3 |
| HXU | 4 | 0 | 12 | 9 |
| Cooling Tower | 4 | 1 | 11 | 12 |
| Pump | 5 | 1 | 20 | 22 |
| Boiler | 6 | 5 | 17 | 19 |
| VAV | 8 | 3 | 30 | 18 |
| CRAC | 10 | 0 | 21 | 28 |
| Chiller | 11 | 0 | 26 | 27 |
| AHU | 55 | 54 | 312 | 172 |

Benchmarking platforms such as OpenCompass (Contributors, 2023) and uncertainty quantification in benchmarks (Ye et al., 2024) enable more reliable evaluation of LLMs across tasks and domains. Studies on knowledge capacity and factuality assessment (Li et al., 2024a; Wei et al., 2024) inform improvements in LLM trustworthiness.

In the tabular and scientific domain, LLMs have shown promise for automatic feature engineering (Han et al., 2024) and scientific knowledge extraction (Bhattacharjee et al., 2024). Efforts to automate dataset updates and maintain evaluation relevance (Ying et al., 2024) address practical challenges in large-scale LLM deployment.

## G    RULE DOCUMENT COLLECTED FROM INDUSTRY EXPERTS.

### G.1    MAINTENANCE RELATED DATASET AND BENCHMARKS

In this appendix, we provide a structured comparison of two closely related datasets: CAMB(Zhang et al., 2025a) and the Wind-Turbine Log benchmark (Malyi et al., 2025) together with our own contribution. While maintenance is a fundamental task in managing industrial physical assets, the challenges addressed by each dataset differ substantially. CAMB focuses on common-sense procedural maintenance knowledge derived from manuals and aviation reference sources. The Wind-Turbine Log dataset targets taxonomy-driven classification of incomplete maintenance records. In contrast, our benchmark centers on automated monitoring and actionable maintenance recommendation, which requires aligning operational conditions with expert-defined rules, integrating sensor data, and supporting temporally grounded reasoning. The comparison given in Table 7 clarifies the unique scope of our work relative to prior datasets.

Table 7: Comparison of CAMB, Wind-Turbine Log, and our benchmark.

| Dimension | CAMB | Wind-Turbine Log | Ours |
|---|---|---|---|
| ArXiv Date | 28 Aug 2025 | 8 Sep 2025 | 19 Sep 2025 (submitted) |
| Dataset Size / Assets | 12 | 1 | 16 |
| Task Type | Common-sense maintenance QA | Log classification | Temporal rule & action mapping |
| Data Sources | Books, manuals, expert docs | Real turbine service logs | Expert rules + real actions from production systems |
| Domain Scope | Aviation | Wind energy (single subsystem) | Industrial data-center |
| Modality | Chinese and English | English | English |
| Construction Process | Internet + books; distractor method not specified | LLM-generated; method details limited | Deterministic, LLM-free pipeline; rule-based, log-driven; difficulty knobs |
| Expert Validation | No | No | Yes |
| Results Summary | Small LLM gaps | - | Significant temporal reasoning difficulty; strong variation across LLMs |

# H  CONDITIONS AS NATURAL LANGUAGE EXPERIMENTS

Table 8: Conditions as Natural Language

| Model | Macro Accuracy | $\Delta$ Macro Accuracy |
|---|---|---|
| claude-3-7-sonnet | 68.22 | -2.58 |
| o1 | 64.53 | -4.33 |
| deepseek-v3 | 62.02 | -6.45 |
| mistral-large | 60.45 | -3.70 |
| qwen2-5-72b-instr. | 56.73 | -3.53 |
| llama-3-3-70b-instr. | 55.16 | -9.62 |
| mistral-medium | 55.89 | -5.09 |
| gemini-1.5-pro | 55.78 | -4.10 |
| mistral-small-3-1. | 53.78 | -6.01 |
| llama-3-1-405b-instr. | 53.81 | -4.37 |
| granite-3-3-8b-instr. | 53.69 | -1.75 |
| gemini-2.0-flash | 49.78 | -6.37 |
| microsoft-phi-4 | 45.64 | -2.73 |
| claude-3-5-haiku | 42.30 | -6.22 |
| llama-3-1-8b-instr. | 39.79 | +1.69 |

Prior work reports that embedding-based models can achieve approximately 66% accuracy on CAMB tasks, particularly in datasets with 7K+ questions covering 12 aircraft types. However, these studies focus primarily on direct QA performance and do not conduct deeper analyses. Specifically, they do not include perturbation analysis (e.g., robustness to noise, paraphrasing, or distractor shifts), do not perform statistical significance testing, and do not evaluate recommendation-oriented models that translate operational conditions into actionable maintenance decisions. As a result, these benchmarks provide valuable baselines but lack the methodological depth necessary to evaluate reasoning robustness, operational reliability, or the transition from question answering to decision-support and recommendation tasks.

# I   GENERALIZABILITY EXPERIMENTS SETUP

**SFT:** For hardware we use 4xNvidia A100 GPUs with 80GB memory. We fine-tune the base model using QLoRA (Dettmers et al., 2023) with FlashAttention-2 (max sequence length $2048$, packed sequences), 4-bit quantization, and LoRA adapters ($r = 16$, $\alpha = 16$), training for 3 epochs with a per-device batch size of 8, a learning rate of $2 \times 10^{-4}$ and a 0.1 warmup ratio.

**GRPO:** For hardware we use 4xNvidia A100 GPUs with 80GB memory. We train for 250 steps, with 16 generations per step and effective batch size of 4 per device. We use Learning Rate (LR) of $5 * 10^{-7}$, Beta of $0.001$, cosine LR scheduler and 0.03 warmup ratio. We use the Hugging Face implementation that excludes prompt length and reward std normalization due to bias Liu et al. (2025).

**Formatting:** For Qwen3-8B we align on a json format with reasoning and answer fields as recommended in the documentation for MCQA questions. For granite-3.3-8B we use <think></think> and <response></response> tags as described in the model card and in llama-3.1-8b we used <think></think> and <answer></answer> tags. Think tags/fields are ommited for SFT. During evaluation, if a model doesn't provide an answer we consider it as wrong.

# J   BACKGROUND: INDUSTRIAL ASSET DIAGNOSTICS SYSTEM

Large corporations and institutions usually own their industrial facilities. Industrial facilities rely on a diverse set of physical assets, including but not limited to chillers, boilers, pumps, compressors, air compressors, and air handling units (AHUs), to maintain safe, efficient, and resilient operations and ensure a smooth working environment. These assets are foundational in sectors like data centers, hospitals, manufacturing plants, and commercial buildings, where equipment failures can lead to operational downtime, safety risks, or financial losses.

Today, to manage these industrial assets proactively, organizations deploy sensor networks that continuously track real-time measurements such as temperature, pressure, flow rate, humidity, and power consumption. Domain experts use this data to define *diagnostic rules*, which map specific combinations of sensor conditions to likely early signals of particular failure modes and ideally recommended follow-up actions, such as inspection, warning bookkeeping, and proactive maintenance. These rules power the early warning systems and suggest the predictive maintenance workflows, allowing operations teams to detect and address potential faults before they escalate to interrupt the operation.

Creating these diagnostic rules is a labor-intensive, highly specialized task. Domain experts must understand the behavior of each asset under various operating conditions, interpret complex sensor relationships, and encode domain knowledge into logical expressions that accurately represent the behavior of each asset. A single facility may require hundreds of rules per asset type or model, each reflecting detailed dependencies between multivariate sensor signals. Rules often involve temporal thresholds, conditional logic, and asset-specific tolerances.

Table 9 presents a set of representative rules derived from production environments, demonstrating the range of conditions and asset behaviors encountered in industrial diagnostics.

This rule-related management domain presents several challenges that complicate automation:

- **High-dimensional, domain-specific sensor data**, often with implicit semantics not found in general corpora.

- **Complex logical dependencies** across multiple sensor conditions, often involving nested boolean logic.

- **Asset heterogeneity**, where similar asset classes behave differently depending on configuration or environment.

- **Tacit expert knowledge** that is rarely documented and typically acquired through experience.

Table 9: Illustrative Examples of Diagnostic and Alert Rules across Industrial Asset Types

| Asset Type | Rule Name | Rule Logic Summary |
|---|---|---|
| AHU | Simultaneous Heating and Cooling | AHU Running; Cooling Valve $\geq$ 5%; Heating Valve $\geq$ 5%; Drain Flags Active; Met for 2 Hours |
| AHU | Heating Valve Open When Warm Outside | AHU Running; OAT - Supply Temp textmore 5°F; Heating Valve textmore 10%; Met for 2 Hours |
| Air Compressor | Pressure Setpoint Attainment | ABS(Pressure - Setpoint) textmore 10 PSI OR Pressure textmore 130 PSI (if setpoint missing); Met for 2 Hours |
| Air Compressor | Flow Flag | Not Monday; Air Flow textmore 120% of Previous Day's Average; Met for 2 Hours |
| Boiler | Excess $O_2$ in Stack | Fuel Flow textmore 5 and Flue Gas $O_2$ exceeds threshold; Met for 2 Hours |
| Boiler | Flue Gas Temperature Setpoint Attainment | Flue Gas Temp below setpoint; Met for 2 Hours |
| Chiller | Temperature Setpoint Attainment | Chiller Running; Supply Temp - Setpoint textmore 5°F; Met for 2 Hours |
| Chiller | Low Supply Temperature | Chiller Running; Setpoint - Supply Temp textmore 3°F; Met for 2 Hours |
| Cooling Tower | Delta T Out of Range | Condenser Return - Supply Temp <5°F; Tower Running; Met for 2 Hours |
| Cooling Tower | Pressure Setpoint Attainment | ABS(Condenser Pressure Diff - Setpoint) textmore 5 PSI; OAT <95°F; Met for 2 Hours |

While diagnostic rules are effective in practice, they do not scale easily. The growing complexity and data richness of industrial systems require tools that can assist in generating, validating, and refining such rules.

Large Language Models (LLMs) offer a promising avenue for this. However, general-purpose LLMs are not trained on sensor semantics or domain-specific diagnostics, and it is unclear whether they can reason over the kinds of multivariate conditions and logic used in real-world maintenance workflows.

Given the domain complexity and the limitations of manual rule engineering, we now present Asset DiagnosticIQ, a benchmark for testing whether LLMs can assist in scalable, high-quality industrial diagnostics.

## K ASSET TYPE DESCRIPTIONS

The asset DiagnosisIQ dataset includes diagnostic rules derived from a wide range of industrial asset types commonly found in commercial buildings, data centers, manufacturing facilities, and other operational environments. Each asset class is associated with domain-specific behaviors, sensor signals, and potential fault conditions that inform the construction of diagnostic questions. This section provides concise descriptions of the primary asset types represented in the dataset, supporting an understanding of their operational roles and diagnostic relevance.

Table 10 provides brief descriptions of the major asset types represented in the Asset DiagnosisIQ dataset. These physical systems are typically monitored via real-time sensor data and are subject to diagnostic rules used for fault detection and predictive maintenance.

Table 10: Industrial Asset Types Addressed in the Diagnostic Rules

| Asset Type | Description |
| --- | --- |
| AHU | Conditions and circulates air as part of an HVAC system. Regulates airflow, temperature, and humidity in commercial buildings. |
| Air Compressor | Converts electrical or mechanical power into pressurized air for pneumatic systems and equipment. |
| Boiler | Heats water or other fluids for use in heating systems, industrial processes, or power generation. |
| Chiller | Removes heat from water using vapor-compression or absorption cycles; supplies chilled water to cooling systems. |
| Cooling Tower | Rejects heat from water-cooled systems by dissipating it into the atmosphere. Common in HVAC and process cooling. |
| CRAC (Computer Room AC) | Cools air in data centers to maintain safe temperature and humidity for IT equipment. Specialized HVAC component. |
| Fan | Drives air movement for ventilation, circulation, or cooling. Includes exhaust, supply, and return fans. |
| Heat Exchanger | Transfers heat between two fluids without mixing. Used for efficient thermal regulation in building systems. |
| Plate & Frame HX | A compact type of heat exchanger using stacked plates to transfer heat between fluid streams. |
| Pump | Moves liquids through mechanical force. Used in chilled water, hot water, and process fluid systems. |
| Terminal Unit | Regulates temperature and air delivery in individual building zones. Includes fan coil units and unit ventilators. |
| VAV (Variable Air Volume Unit) | Controls airflow to a zone by varying damper position, often part of demand-driven ventilation. |
| Condenser | Rejects heat from refrigerant cycles in chillers or heat pumps. Includes air- or water-cooled variants. |
| ERV (Energy Recovery Ventilator) | Transfers heat and moisture between exhaust and incoming fresh air streams to improve HVAC efficiency. |
| Water Heater | Provides domestic or process hot water, separate from large-scale boiler systems. May be gas or electric. |
| UPS (Uninterruptible Power Supply) | Supplies provides temporary backup power during grid interruptions to protect critical equipment. |
| Electrical Panel | Distributes power to facility circuits and equipment; may be monitored for load balancing or safety. |

## L  CASE STUDY: CLOSED-LOOP WATER-COOLING CHILLER WITH COOLING TOWER

This case study illustrates a realistic multi-component diagnostic scenario for a **closed-loop water-cooled chiller system** paired with a **cooling tower**. It demonstrates how expert-authored rules, time-persistent sensor conditions, and actionable maintenance logic can be represented in the DiagnosticIQ framework. The case highlights cross-asset, multi-sensor reasoning a primary challenge captured by the `DiagnosisIQPro` dataset.

Figure 7 illustrates the schematic layout of a closed-loop water-cooled chiller system. The diagram captures the three key subsystems:

- **Refrigerant Loop:** Evaporator, compressor, condenser, and expansion valve arranged in a standard vapor-compression cycle.
- **Chilled Water Loop:** Chilled water pump circulates through the building's cooling coils and returns to the evaporator.

- **Condenser Water Loop:** Removes heat from the condenser and rejects it to the atmosphere via a cooling tower.

The Building Automation System (BAS) coordinates the entire process by issuing control signals to the compressor, pumps, and other critical components.

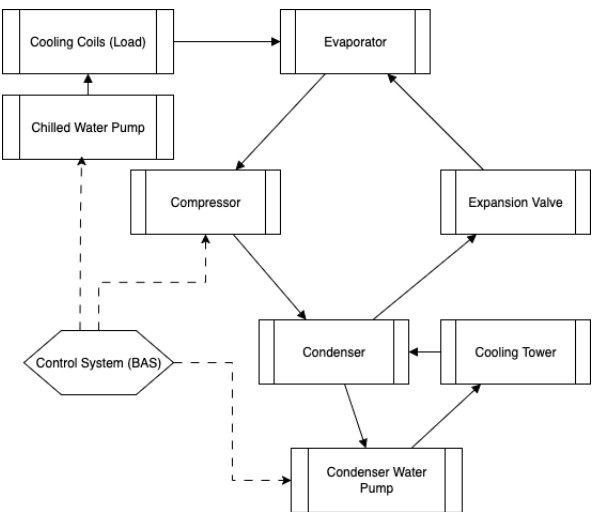

Figure 7: Schematic of a closed-loop water-cooled chiller system

### L.1 SYSTEM CONTEXT AND MOTIVATION

Water-cooled chillers are used in high-performance HVAC systems where condenser heat is rejected via cooling towers. These systems rely on evaporators, compressors, condenser loops, and building automation systems (BAS) to coordinate thermal transfer. Rule-based diagnostics are essential for early detection of inefficiencies or faults, particularly in critical environments like data centers and hospitals. As shown in Table 11, each component is associated with a distinct set of sensors and diagnostic KPIs.

### L.2 COMPONENT-LEVEL DIAGNOSTIC RULES

We present a curated set of expert-authored diagnostic rules, each associated with a chiller subsystem and defined by time-persistent Boolean logic. Table 12 summarizes selected rules aligned with the DiagnosticIQ schema.

### L.3 INTEGRATED DIAGNOSTIC SCENARIO

This case demonstrates cross-asset rule activation. The observed conditions span the chiller and cooling tower, requiring compositional reasoning. In the following, The meaning of OAT is Outside Air Temperature.

**Observed Conditions (QC):**

- Chiller Running
- Supply Temp - Setpoint Temp $> 5°F$
- Evaporator $\Delta T < 7°F$
- Condenser Water Return - Supply Temp $< 5°F$
- Cooling Tower Supply Temp $<$ Setpoint - $4°F$
- OAT $> 43°F$

**Activated Rules ($T_{R_i}$):**

Table 11: Components, Sensors, and KPIs in a Closed-Loop Water-Cooled Chiller System with Cooling Tower

| Component | Common Sensors / Meters | Associated KPIs | Purpose / Insight |
|---|---|---|---|
| Evaporator | Inlet Temp, Outlet Temp, Water Flow | $\Delta T_{\text{Evap}} = T_{\text{in}} - T_{\text{out}}$ Cooling Load $= \dot{m} \cdot c_p \cdot \Delta T$ | Measures heat absorbed from chilled water; identifies underperformance or fouling. |
| Compressor | Power (kW), Amps, Suction/Discharge Pressure, Vibration | Compressor Efficiency = Cooling Load / Power Compression Ratio $= P_{\text{dis}}/P_{\text{suc}}$ | Assesses mechanical load, efficiency, early signs of failure or degradation. |
| Condenser | Inlet Temp, Outlet Temp, Water Flow | $\Delta T_{\text{Cond}} = T_{\text{in}} - T_{\text{out}}$ Approach Temp $= T_{\text{refrigerant}} - T_{\text{cond-out}}$ | Evaluates heat rejection quality; detects fouling, flow loss, scaling. |
| Expansion Valve | Pre/Post Temp, Pressure | Superheat / Subcooling Temperatures | Indicates refrigerant charge level, valve responsiveness, control precision. |
| Pump | Flow Rate, $\Delta P$ (Suction-Discharge), Power, Status | Pump Efficiency = Flow / Power $\Delta P$ Stability | Verifies circulation; detects pump wear, airlocks, or blockage. |
| Cooling Tower | Return Temp (from condenser), Supply Temp (to condenser), Fan Speed, OAT, Water Level | $\Delta T_{\text{Tower}} = T_{\text{return}} - T_{\text{supply}}$ Approach Temp $= T_{\text{supply}} - T_{\text{ambient}}$ | Validates heat rejection; detects overcooling, bypass issues, fan control faults. |
| Control System | Setpoints, Run Status, Fault Logs, Mode Indicators | Setpoint Attainment $= \lvert T_{\text{supply}} - T_{\text{setpoint}} \rvert$ Cycle Count, Runtime Logs | Supports alarm generation, diagnostics of control tuning, overshoot, inefficiency. |

- Chiller - Setpoint Not Met

- Chiller - Cooling $\Delta$T Low

- Cooling Tower - Water Too Cold

**Observation** ($O_i$): Cooling tower is overcooling condenser water, reducing head pressure and impairing chiller performance.

**Action** ($A_i$): Tune tower fan control logic or enable bypass to raise condenser water temperature.

This case study demonstrates the type of multivariate, cross-component diagnostic reasoning supported by the `DiagnosisIQPro` dataset. By grounding rule activation in realistic sensor logic and translating conditions into actionable maintenance decisions, the example highlights the practical value of structured condition-action QA benchmarks for industrial asset management.

## M  DIAGNOSTIC RULE LOGIC CATEGORIZATION WITH EXAMPLES

This section presents a structured classification of diagnostic rules based on their underlying Boolean logic. Diagnostic rules are widely used in automated fault detection and energy analytics systems to evaluate sensor data from building systems such as air handling units (AHUs), chillers, boilers, and compressors. Understanding the logical structure of these rules enhances interpretability, facilitates rule development, and supports systematic debugging. Tables of 13 and 14 outline common logic categories and provide real-world examples to illustrate each structure.

Table 12: Sample Diagnostic Rules for Closed-Loop Water-Cooled Chiller System

| Component | Rule Name | Associated Sensors / KPIs | Condition Logic Summary |
|---|---|---|---|
| Control System | Cooling Temp Setpoint Not Met | Supply Temp, Setpoint Temp (KPI: $\Delta T_{\text{Setpoint}}$) | Chiller Running $\wedge$ (Supply Temp - Setpoint Temp $> 5°$F) for 2 hrs |
| Control System | Low Supply Temperature | Supply Temp, Setpoint Temp (KPI: $\Delta T_{\text{Setpoint}}$) | Chiller Running $\wedge$ (Setpoint - Supply Temp $> 3°$F) for 2 hrs |
| Evaporator | Cooling $\Delta T$ Low | Return Temp, Supply Temp, OAT (KPI: $\Delta T_{\text{Evap}}$) | Chiller Running $\wedge$ P&F Off $\wedge$ $\Delta T < 7°$F $\wedge$ OAT $> 37°$F for 4 hrs |
| Compressor | Efficiency Exceeds Threshold | Power Input, Cooling Load (KPI: Chiller Efficiency) | Chiller Running $\wedge$ Efficiency $>$ design parameter for 2 hrs |
| Condenser | Flow Detected While Off | Condenser Water Flow, Run Status | Chiller Off $\wedge$ Flow $> 50$ GPM for 2 hrs |
| Compressor | Load Low | Load %, Amps, Full Load Amps | Chiller Running $\wedge$ (Load % $< 30\%$ $\vee$ Amps / FLA $< 30\%$) for 2 hrs |
| Evaporator | Evaporator Approach High | Supply Temp, Refrigerant Temp (KPI: Evap Approach) | Chiller Running $\wedge$ (Supply - Refrigerant Temp $> 4°$F) for 3 hrs |
| Condenser | Condenser Approach High | Liquid Temp, Return Temp (KPI: Cond Approach) | Chiller Running $\wedge$ (Condensate - Return Temp $> 4°$F) for 3 hrs |
| Pump / Control | $\Delta P$ Not at Setpoint | $\Delta P$, $\Delta P_{\text{Setpoint}}$ | ABS($\Delta P$ - $\Delta P_{\text{Setpoint}}$) $> 4$ PSI for 3 hrs |
| Control System | Excessive Power While Off | Power, Run Status | Chiller Off $\wedge$ Power $> 5$ kW for 3 hrs |
| Control System | Chiller Cycling | Run Status Log | Status changed $\geq 4$ times in 8 hrs |

Table 13: Boolean Logic Categories Used in Diagnostic Rules

| Logic Category | Explanation |
|---|---|
| Conjunctive (AND) | All listed conditions must be simultaneously satisfied. Example pattern: $c_1 \wedge c_2 \wedge \cdots \wedge c_n$ |
| Disjunctive (OR) | Any one condition is sufficient to trigger the rule. Example pattern: $c_1 \vee c_2 \vee \cdots \vee c_n$ |
| Mixed (AND-OR) | Structured combinations of conjunctive and disjunctive logic, typically in disjunctive normal form (DNF). Example pattern: $(c_1 \wedge c_2) \vee (c_3 \wedge c_4)$ |
| Negation-based | Includes explicitly negated conditions. Example pattern: $c_1 \wedge \neg c_2$ or $\neg(c_1 \vee c_2)$ |

Categorizing diagnostic rules by their logic structure enables clearer reasoning about their behavior, performance, and possible interactions. It also aids in identifying redundant or conflicting rules in complex control systems. As building analytics platforms scale, such structured representations provide a foundation for more advanced techniques like rule validation, automated rule generation, and explainable diagnostics.

## N DATASET GENERATION METHODOLOGY

**Algorithm Description:** The QA Generation Pipeline (See Algorithm 1) takes as input a set of expert-defined rules $\{\mathcal{R}^i\}$, asset descriptions (Desc), and a parameter max_n_choices controlling the maximum number of answer options per question. For each rule, the pipeline extracts atomic

Table 14: Examples of Diagnostic Rules Categorized by Logical Structure.

| Asset Name/Rule | Logic Category | Logical Expression Summary |
|---|---|---|
| Air Compressor - Pressure Setpoint Attainment | Disjunctive (OR) | (ABS[Pressure – Setpoint] > 10 PSI OR Pressure > 130 PSI) |
| AHU - Simultaneous Heating and Cooling | Mixed (AND-OR) | AHU Running AND (Cooling Valve $\geq$ 5% OR Preheat Valve $\geq$ 5%) AND (Drain Flags = 0) |
| AHU - Heating Valve Open when Warm Outside | Mixed (AND-OR) | AHU Running AND (OAT – SAT > 5°F OR SAT Not Reporting) AND (Heating Valve > 10% OR Preheat Valve > 10%) |
| Boiler - Excess O$_2$ in Stack | Disjunctive (OR) | (Gas Flow > 5 AND Flue O$_2$% > threshold OR Flue O$_2$% > threshold IF Fuel Flow Not Reporting) |
| CRAC - Limited Cooling Warning | Conjunctive (AND) | CRAC Running AND (Return Temp $\leq$ Supply Temp + 3°F) |
| Chiller - Cooling Substance Temperature Setpoint Attainment | Conjunctive (AND) | Chiller Running AND (Supply Temp – Setpoint > 5°F) |

conditions from the condition tree $\mathcal{TR}^i$, retrieves the corresponding asset description, and generates question-answer pairs by selecting and eliminating candidate observations using similarity metrics and heuristics. It then combines each extracted condition with all relevant question-option-answer tuples to build the final dataset $DS_\mathcal{Q}$, facilitating systematic benchmarking of maintenance action recommendations.

---

**Algorithm 1** QA Generation Pipeline

---

**Input:** $\{\mathcal{R}^1, \ldots, \mathcal{R}^{N_\mathcal{R}}\}$, $Desc$, $max\_n\_choices$
**Output:** $DS_\mathcal{Q}$

1: Initialize $DS_\mathcal{Q} \leftarrow []$
2: **for** each $\mathcal{R}^i \in \{\mathcal{R}^1, \ldots, \mathcal{R}^{N_\mathcal{R}}\}$ **do**
3:      $\{QC_j^i\}_{j=1}^{N_{cond}} \leftarrow extracted\_conditions(\mathcal{TR}^i)$
4:      $AD^i \leftarrow get\_asset\_desc(\mathcal{R}^i, Desc)$
5:      $\{(QP_j^i, OPT_j^i, A_j^i)\}_{j=1}^{N_{sel}} \leftarrow extracted\_obs\_sel(\mathcal{R}^i, \alpha, RRSim, UO)$
6:      $\{(QP_j^i, OPT_j^i, A_j^i)\}_{j=1}^{N_{eli}} \leftarrow extracted\_obs\_eli(\mathcal{R}^i, \beta, RRSim, UO)$
7:      $all\_opts \leftarrow \{(QP_j^i, OPT_j^i, A_j^i)\}_{j=1}^{N_{sel}} \cup \{(QP_j^i, OPT_j^i, A_j^i)\}_{j=1}^{N_{eli}}$
8:      **for** each $QC_{j1}^i \in \{QC_j^i\}_{j=1}^{N_{cond}}$ **do**
9:          **for** each $(QP_{j2}^i, OPT_{j2}^i, A_{j2}^i) \in all\_opts$ **do**
10:              $\mathcal{Q}^i \leftarrow (AD^i, QC_{j1}^i, QP_{j2}^i, OPT_{j2}^i, A_{j2}^i)$
11:              Append $\mathcal{Q}^i$ to $DS_\mathcal{Q}$
12:          **end for**
13:      **end for**
14: **end for**
15: **return** $DS_\mathcal{Q}$

---

### N.1 RULE TO RULE SIMILARITY MAP

We utilize Rule to Rule Similarity **RRSim** mapping during the creation of dataset. The similarity is calculated by initially embedding the text components (*asset_type*, *conditions*) of each rule using a all-mpnet-base-v2 embedding model. Then the embedding to embedding similarity is calculated according to cosine similarity.

## N.2 DIAGNOSISIQPRO

The **DiagnosisIQPro** dataset extends *DiagnosisIQ* to evaluate model performance under more challenging conditions with larger option sets. To enable direct comparison, for each question $\mathcal{Q}^i$ in *DiagnosisIQ*, we retain the asset description $AD^i$, observed conditions $QC^i$, question prompt $QP^i$, and ground-truth answer $A^i$, while expanding the set of answer options $OPT^i$ by adding additional plausible but incorrect choices.

For *selection*-type questions, we increase the number of incorrect options by resampling from observations of rules that are semantically similar yet distinct, ensuring that distractors remain relevant but incorrect. For *elimination*-type questions, we similarly augment the sets of correct and incorrect options to increase task complexity, leveraging domain-informed similarity measures (e.g., **RRSim**) to maintain logical coherence.

This augmentation more closely mimics real-world industrial scenarios, where practitioners must consider numerous potential failure causes, thereby testing the robustness and discriminative capabilities of language models in high-option environments.

## N.3 DIAGNOSTICIQPERT

We create a perturbation dataset DiagnosisIQPert to analyze the sensitivity of model responses to minor variations in the questions. This dataset is derived by manipulating DiagnosisIQ questions through several transformations: randomly shuffling the order of conditions and options, adding parentheses around option labels (e.g., $A \rightarrow (A)$), changing option labels (e.g., $A, B, C \rightarrow P, Q, R$), and substituting one question prompt $QP^i$ with another. These perturbations help assess the robustness and consistency of language models when faced with slight changes in input formatting or phrasing.

Figure 8: Example Question

```
Please select the correct option(s) from the following options given the question:
Question:
## Asset Description:
AHU: Air Handling Unit: A device used to condition and
circulate air as part of a heating, ventilating, and air-conditioning (HVAC) system.

## Conditions:
- AHU Running
- Outside Air Damper % < 15% AND Outside Air Damper Minimum % Not  Reporting
- Economizer Mode AND Supply Relative Humidity % Not Reporting
- OAT < Setpoint Temperature
- Outside Air Damper %
- OAT > 37 °F
- Outside Air Damper % Does NOT = Daily Average
- SubType NOT OAU, RAS, RAU

## How long the conditions were met:
Met for 2 Hours

Looking at the current state of the asset, what is the MOST likely cause among the
options?

Options:
(P) Control system sent the wrong command
(Q) Belts are loose or broken
(S) Broken Belt
(R) Vanes at wrong angle
Your output must strictly follow this format:
{"answer": <the list of selected options, e.g., ["(P)", "(R)"]>}
```

## N.4 DIAGNOSTICIQVERBOSE

To assess the symbolic understanding in the context of maintainance action recommendation we create a variant of DiagnosisIQ questions named DiagnosticIQVerbose. To create this we initially embedded the conditions of each question using a all-mpnet-base-v2 embedding model. Then the embedding are cluster according to cosine similarity to get 10 representative questions representing the cluster groups. We manually convert the conditions into natural language of these questions and then use these as in-context examples and prompt a mistral-large to generate the natural language representation for the rest of the questions.

```
Your task is to read the asset description (## Asset Description:) and conditions (##
    Conditions:) applied on the asset and
write the conditions (## Conditions:) in natural language several examples are provided,
    complete the last sample.

## Conditions:
AHU Running
OAT < 80F
Cooling Valve % > 97%
ABS(Supply Air Temperature Setpoint - Supply Air Temperature) > 3IF Setpoint Reporting

## Conditions in Natural Language:
The asset is running while the outside temperatue is less than 80Fahrenheit and the
    units cooling valve is nearly fully open ( Cooling Valve is open more than 97% )
and further tha absolute value of the difference between set threshold of air
    temperature and supply air temperature is greater than 3
Fahrenheit

...

## Asset Description:
AHU: Air Handling Unit: A device used to condition and circulate air as part of a
    heating, ventilating, and air-conditioning (HVAC) system.

## Conditions:
AHU Running
OAT > 35F
Preheat Valve % > 97%

## Conditions in Natural Language:
```

Listing 2: Prompt used to convert symbolic representation natural language

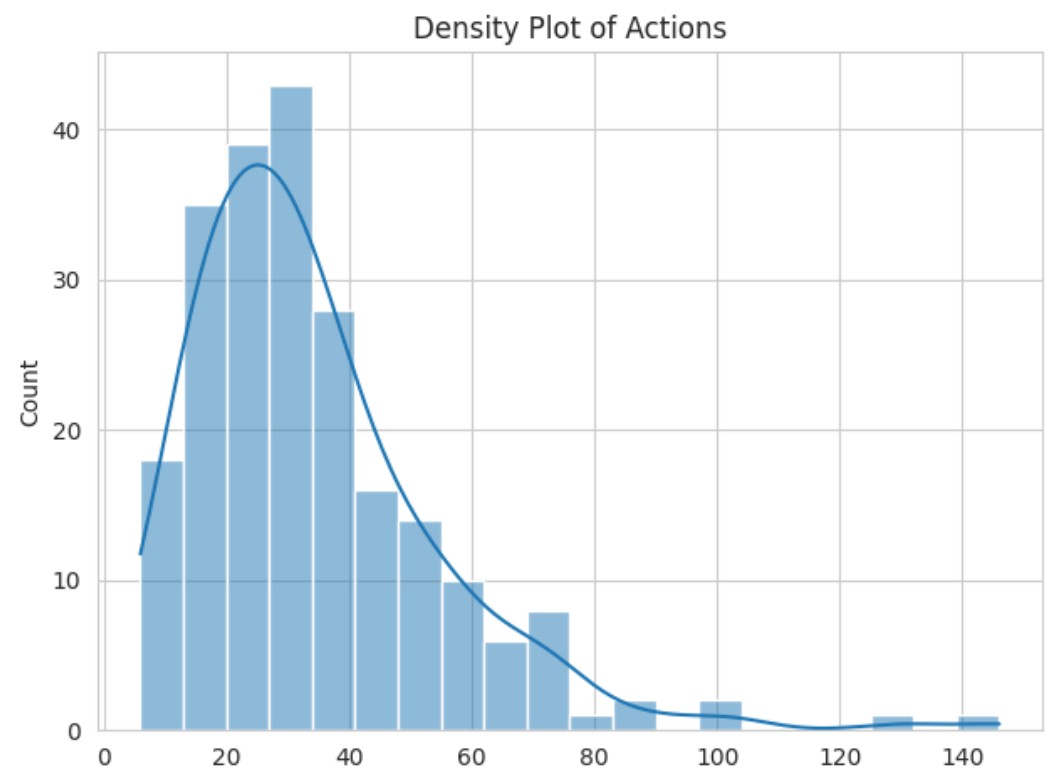

Figure 9: The word count distribution of unique actions in the expert curated dataset

## O    MAINTENANCE ACTION RECOMMENDATION ENGINE

The Maintenance Action Recommendation Engine takes a semi-defined rule as input in the form $(asset\_type, \mathcal{TR}^i, t^i)$, and returns a set of recommended maintenance actions $\mathcal{O}^i$ corresponding to a given rule $\mathcal{R}^i$.

We begin by selecting all possible actions applicable to a specific asset, resulting in a large action space $AS$. These actions are sourced from the collection of unique operations defined in our Expert-Curated Rule Documents (see Sec. 3.1).

To manage the scale of this action space, we adopt a divide-and-conquer strategy by chunking the actions into manageable segments, as described in Algorithm 2.

we utilize the DiagnosticIQ as a database of QA to inject examples giving the question and the answer rather

An example of a prompt generated by the get_llm_prompt function is shown in Fig. 10.

## P    EXAMPLES OF LLM RATIONALE

This section provides an examples of rationales that were generated for the purposes of evaluating LLM domain understanding and reasoning.

### P.1    LLM AND ANNOTATOR DISAGREEMENTS.

We notice that experts do not always agree with certain rationale which can be seen as outliers in the Fig 5. We present such an example and the reasoning given by the expert to give a low rating. The rationale that was generated and the cosponsoring expert comment is denoted in Fig 12

Figure 10: MAReE Prompt with 1 example question.

```
## Asset Description:
Cooling Tower
## Conditions:
- Cooling Tower Running
- 55 degF < Outside Air Temp < 80 degF
- Supply Temp Setpoint = Previous Hour Supply Air Temp Setpoint
- Supply Temp Setpoint = Previous Daily Average Supply Air Temp Setpoint
## How long the conditions were met:
Met for 3 Hours Checking Previous 3 Days Daily Average
Analyse the given conditions of the presented asset and rank
the options that MOST likely gives the reason
for the conditions?
A. Outside air temperature sensor failure
B. Fans are off
C. Check fans and condenser water pumps
D. VFD operation
E. Too many cnodenser pumps running
F. Logic issues for the cooling tower
G. Cooling tower reset in manual
H. Fan is overridden
I. Too few cooling towers running
J. Static pressure sensors need calibration, repair or replacement
## Use following Questions and answers as help for the ranking.
### Example 1
### Asset Description:
Cooling Tower: A heat rejection device that cools water or other fluids
by transferring heat to the atmosphere. It is commonly used in HVAC systems,
power plants, and industrial processes.
### Conditions:
- Cooling Tower { Condenser Water is too cold
- Cooling Tower Running
- OAT > 43 °F
- Condenser Water Supply Temperature to Chiller < 55 °F
IF Condenser Water Temperature Setpoint NOT Reporting
### How long the conditions were met:
Met for 2 Hours
Review the listed conditions and identify which option MOST accurately accounts for
them.
A. Fan blades at incorrect pitch
B. Load is too low or fluctuates
C. Fan is overridden
D. If unit resets based on VAV damper position exempt from this rule.
Answer: C. Fan is overridden
Your output must strictly follow this format:
{"option": <list of the option tag e.g. ['A', 'B', 'C', 'D', 'E']>,
"score":<list of scoring value inline with rank ranging from 1,-1 eg: [1.0, 0.9, 0.8,
0.7, 0.6]>,
"rank":<list of the rank eg: [10, 9, 8, 7, 6]>}

Your output in a single line:
```

## P.2 LLM AND ANNOTATOR AGREEMENTS.

On the flip side we provide an example where the annotators agree with the LLM generated rationale
in 13

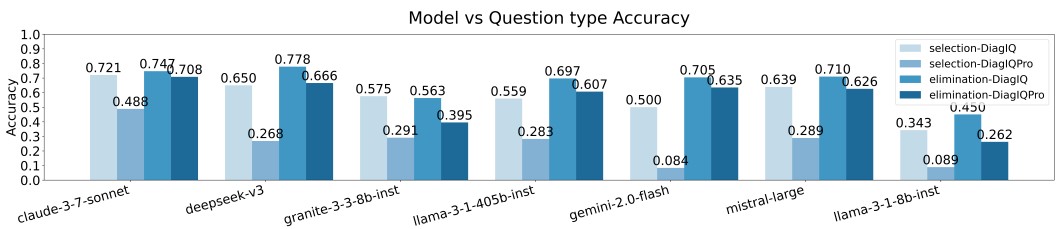

Figure 11: Model vs Question type Accuracy

---

**Algorithm 2** Maintenance Action Recommendation Engine

---

**Input:** $DS_{\mathcal{Q}}$, $asset\_type$, $\mathcal{R}^i$, $N_{obs}$, $top_k$
**Output:** $\mathcal{O}$
  1: **function** SELECTACTIONS($\mathcal{Q}_{list}$, $\mathcal{R}$, $obs$, $k$)
  2:     $\mathcal{Q}_{sel} \leftarrow$ GET_SIMILAR_QUESTIONS($\mathcal{Q}_{list}$)
  3:     $prompt \leftarrow$ GET_LLM_PROMPT($\mathcal{R}, obs, \mathcal{Q}_{sel}$)
  4:     $order \leftarrow$ LLM_ANSWER($prompt$)
  5:     Re-order $obs$ using $order$
  6:     $sel\_obs \leftarrow$ SELECT_TOP_OBSERVATIONS($obs, k$)
  7:     **return** $sel\_obs$
  8: **end function**
  9: **function** DYNAMICACTIONRANKING($\mathcal{Q}_{list}$, $Actions$, $asset\_type$, $\mathcal{R}^i$, $N_{obs}$, $top_k$)
 10:     $sel\_obs_{all} \leftarrow [\ ]$
 11:     **for** $i \leftarrow 0$ **to** $|Actions| - 1$ **step** $N_{obs}$ **do**
 12:         $list\_obs \leftarrow Actions[i : i + N_{obs}]$
 13:         $sel\_obs \leftarrow$ SELECTACTIONS($\mathcal{Q}_{list}, \mathcal{R}^i, list\_obs, top_k$)
 14:         Append $sel\_obs$ to $sel\_obs_{all}$
 15:     **end for**
 16:     **return** DYNAMICACTIONRANKING($\mathcal{Q}_{list}$, $sel\_obs_{all}, asset\_type, \mathcal{R}^i, N_{obs}, top_k$)
 17: **end function**
 18: $AS \leftarrow$ GET_ASSET_ACTION_SPACE($asset\_type$)
 19: $\mathcal{O} \leftarrow$ DYNAMICACTIONRANKING($DS_{\mathcal{Q}}, AS, asset\_type, \mathcal{R}^i, N_{obs}, top_k$)
 20: **return** $\mathcal{O}$

---

## Q    VARIANCE ACROSS MODEL FAMILIES

Table 15 and Table 16 report the average set size of predictions for the DiagnosticIQ and Pro datasets, respectively. Across both datasets, models consistently exhibit higher average scores on incorrectly answered items compared to the overall item set, indicating systematic patterns in error severity. Results from both the two-sample t-test and the Mann–Whitney U test yield extremely small p-values for nearly all models, confirming that the score distributions for correct and incorrect items differ significantly. These findings suggest that models not only struggle more with difficult cases but also produce substantially larger deviations when they fail, reflecting a meaningful gap in model calibration under challenging conditions.

## R    SEMANTIC RANK ANALYSIS.

To analyze the nature of model errors in the MCQA task, we computed a *semantic rank* for each incorrectly predicted option. Each answer choice was first embedded into a semantic vector space using a pretrained embedding model. For every wrong prediction, we measured its distance to the correct option and ranked all options from closest to farthest. A semantic rank of 1 indicates that the model selected the wrong option that is most semantically similar to the correct answer ("perfect semantic confusion"), while higher ranks indicate progressively less semantically related choices. We then report the mean and standard deviation of these ranks for each model and dataset, providing

Figure 12: Rationale Generated where annotator disagreed.

```
### Asset Description:
Pump: A device used to move fluid or gas from one place to another.
It can include various types of pumps, such as centrifugal pumps,
positive displacement pumps, and diaphragm pumps.

### Conditions:
- Pump Running
- SubType NOT HW
- Pump Motor VFD Output % > 35%
- Pump Motor VFD Output % = Daily Average Pump Motor VFD Output %
- OAT < 95 °F
- ABS(Pump Differential Pressure - Pump Differential Pressure Setpoint) > 4psi
IF Reporting

### How long the conditions were met:
Met for 2 Hours

Analyse the given conditions of the presented asset and select the option that
MOST likely gives the reason for the conditions?
A. Lighting relay schedule over written
B. Mis-sized equipment
C. Unit bypassing or blowing off too much air
D. Static pressure sensors need calibration, repair or replacement

# Answer

Answer: D. Static pressure sensors need calibration, repair or replacement

Guidance Rationale: Given that the pump is running and the motor's variable
frequency drive (VFD) output percentage is both above 35% and equal to the
daily average, this indicates that the pump is operating under stable and
expected load
conditions. However, there is a significant deviation in the pump differential
pressure from its setpoint, exceeding 4 psi, which points to a potential issue
in how pressure is being
measured. Since the outside air temperature is below 95°F, it is unlikely that
environmental conditions are responsible for the discrepancy.

Therefore, the most likely explanation is that the static pressure sensors
require calibration, repair, or replacement,to ensure accurate readings and
efficient pump operation.
```

```
Expert Rating
2

    Expert Comment
    This rule is looking for a VFD that should be changing speed but is not.  The
    VFD or control system may be in manual mode or there is a control system
    failure.  The reference  positive displacement pumps and diaphragm pumps does
    not make sense in the context of this rule.
```

a quantitative measure of whether model mistakes are near-miss semantic errors or more distant, less meaningful deviations.

Table 17 shows representative semantic ranks for selected models in both simple and complex MCQA datasets.

Figure 13: Rationale Generated where annotator disagreed.

```
### Asset Description:
Chiller: A device that removes heat from a liquid through a vapor-compression
or vapor-absorption refrigeration process. The cooled liquid can then be used
to cool equipment, buildings, or other systems.

### Conditions:
- Run Status = 0
- Power Input > 5 KW

### How long the conditions were met:
Met for 3 Hours

What underlying factor, as presented in the options, could explain the current
asset state?
A. Excessive Cooling
B. Could be oil heater
C. Filters clogged
D. Excess equipment is on

# Answer

Answer: B. Could be oil heater

Guidance Rationale:
Given that the chiller's run status is 0, it is not actively performing its
primary cooling function. However, the system is still consuming more than 5 kW
of power for a continuous duration of 3 hours, indicating non-trivial energy
usage during downtime. This suggests that the power draw is likely attributed to
an auxiliary component, rather than the main compressor or cooling cycle. A
common explanation for such behavior is the operation of an oil heater, which is
designed to maintain the proper viscosity of lubricants or ensure safe startup
during idle periods. This aligns with the observation that the chiller is
powered but not running, making the presence of an oil heater the most plausible
cause.

In contrast, excessive cooling (Option A) would imply the chiller is
running, which contradicts the run status. Clogged filters (Option C) typically
affect efficiency rather than causing significant power draw during downtime.
Excess equipment being on (Option D) is less likely as it would imply additional
loads unrelated to the chiller's internal components, which is not supported by
the given conditions. Therefore, the most reasonable explanation is "Could be
oil heater."
```

```
Expert Rating
10

Expert Comment
I liked that it also (maybe unintentionally) hinted at the fact that it could
be D because of an auxiliary component being on.
```

Table 15: Models sorted by Avg_Set_All (ascending) for Diag IQ dataset.

| model_id | avg_set_all | avg_set_wrong | t_p_value | mannwhitney_p_value |
|---|---|---|---|---|
| granite-3-3-8b-instruct | 1.04339 | 1.10413 | 2.15744e-06 | 3.38142e-36 |
| claude-3-7-sonnet | 1.05112 | 1.18699 | 1.23794e-51 | 1.14479e-138 |
| llama-4-maverick | 1.05979 | 1.18059 | 6.02828e-82 | 1.90411e-171 |
| gemini-1.5-pro | 1.08655 | 1.25043 | 1.71241e-73 | 8.96014e-142 |
| o1-new | 1.08744 | 1.29893 | 2.3592e-107 | 8.6181e-271 |
| o1 | 1.09013 | 1.30271 | 2.32491e-107 | 1.06305e-262 |
| deepseek-v3-h200 | 1.11584 | 1.36080 | 1.52242e-140 | 2.88267e-304 |
| gpt-5-2025-08-07 | 1.12377 | 1.38422 | 7.22468e-161 | 0 |
| gemini-2.5-pro | 1.18984 | 1.51922 | 2.59789e-217 | 0 |
| llama-3-3-70b-instruct | 1.19536 | 1.49302 | 1.10683e-283 | 0 |
| mistral-large | 1.21928 | 1.63589 | 2.83552e-224 | 0 |
| qwen2-5-72b-instruct | 1.22676 | 1.61442 | 0 | 0 |
| mistral-small | 1.23513 | 1.59002 | 3.66404e-298 | 0 |
| llama-3-1-405b | 1.23711 | 1.57883 | 0 | 0 |
| llama-3-1-8b-instruct | 1.28406 | 1.45728 | 1.69264e-235 | 2.61701e-178 |
| mistral-medium-2505 | 1.28744 | 1.74535 | 0 | 0 |
| microsoft-phi-4 | 1.29849 | 1.57488 | 0 | 0 |
| gemini-2.0-flash | 1.30179 | 1.66524 | 0 | 0 |
| Qwen3-8B | 1.43513 | 1.76889 | 0 | 0 |
| claude-3-5-haiku | 1.46637 | 1.83961 | 0 | 0 |

Table 16: Models sorted by Avg_Set_All (ascending) for Diag IQ Pro dataset.

| model_id | avg_set_all | avg_set_wrong | t_p_value | mannwhitney_p_value |
|---|---|---|---|---|
| granite-3-3-8b-instruct | 1.22911 | 1.35481 | 1.71521e-115 | 3.46688e-110 |
| claude-3-7-sonnet | 1.24439 | 1.52896 | 1.10791e-197 | 3.35434e-288 |
| gpt-5-2025-08-07 | 1.73214 | 2.22818 | 0 | 0 |
| deepseek-v3-h200 | 1.78954 | 2.22980 | 0 | 0 |
| llama-3-3-70b-instruct | 1.80590 | 2.19058 | 0 | 0 |
| llama-3-1-405b | 1.82952 | 2.28790 | 0 | 0 |
| gemini-2.5-pro | 1.84101 | 2.37583 | 0 | 0 |
| gemini-1.5-pro | 1.84410 | 2.16867 | 0 | 0 |
| microsoft-phi-4 | 1.92840 | 2.22481 | 0 | 0 |
| qwen2-5-72b | 1.98206 | 2.46423 | 0 | 0 |
| claude-4-sonnet | 2.04499 | 2.55945 | 0 | 0 |
| llama-3-1-8b-instruct | 2.21933 | 2.41447 | 0 | 2.01685e-306 |
| mistral-large | 2.16280 | 2.83141 | 0 | 0 |
| mistral-small | 2.35321 | 2.89037 | 0 | 0 |
| Qwen3-8B | 2.36383 | 2.59790 | 0 | 0 |
| gemini-2.0-flash | 2.44664 | 2.82707 | 0 | 0 |
| mistral-medium-2505 | 2.49567 | 3.14170 | 0 | 0 |
| claude-3-5-haiku | 2.49671 | 2.77221 | 0 | 0 |
| o1 | 2.56801 | 3.12219 | 0 | 0 |

Table 17: Representative semantic-rank scores of incorrect predictions for simple and complex MCQA datasets. Rank = 1 corresponds to perfect semantic confusion; higher values indicate farther semantic deviation.

| Model | Dataset | Mean Rank | Std Dev |
|---|---|---|---|
| Claude-4 Sonnet | Simple | 2.71 | 0.92 |
| LLaMA-3-1-8B | Simple | 2.81 | 0.76 |
| Gemini-2.5 Pro | Simple | 2.87 | 0.80 |
| GPT-5 (Aug-2025) | Simple | 2.84 | 0.77 |
| Claude-4 Sonnet | Complex | 4.80 | 2.70 |
| LLaMA-3-3-70B | Complex | 4.99 | 2.54 |
| Mistral-Large | Complex | 4.98 | 2.60 |
| Gemini-2.0 Flash | Complex | 5.28 | 2.62 |

