# OpenReview forum: "Augmenting Industrial Maintenance with LLMs: A Benchmark, Analysis, and Generalization Study"
_ICLR.cc/2026/Conference — Submitted to ICLR 2026_

### Official Review · Reviewer_Lrjy · 2025-10-31

**Soundness:** 2
**Presentation:** 3
**Contribution:** 3
**Rating:** 2
**Confidence:** 2

**Summary:**

The paper introduces DiagnosticIQ, a benchmark for evaluating whether LLMs can recommend maintenance actions from symbolic, time-persistent sensor conditions used in industrial monitoring. The dataset contains 6,690 MCQs drawn from 120 rules across about 16 asset types, with several variants: DiagnosticIQPro, DiagnosticIQPert, DiagnosticIQRationale, and DiagnosticIQVerbose. The authors evaluate 15 LLMs in a zero-shot setting and report a leaderboard. Macro accuracy is highest for Claude-3-7-Sonnet, and most models drop sharply on the Pro split with larger answer choices. They also present a small human study assessing model rationales, a cross-asset fine-tuning study with SFT and GRPO, and detailed analyses by asset and by question type.

**Strengths:**

- The paper targets a real gap: connecting anomaly rules to actionable maintenance guidance at scale.
- The rules-to-MCQ formulation is well motivated by real maintenance workflows.
- The benchmark construction pipeline is transparent and reproducible: condition trees are converted to disjunctive normal form, and question types are systematically constructed.
- It covers diverse evaluation axes, including robustness to prompt perturbations, per-asset performance, question-type differences, and cross-asset transfer with SFT and GRPO.

**Weaknesses:**

- Many models on the leaderboard seem outdated and inconsistent. There are more recent closed-source models for Gemini and OpenAI's reasoning models than the ones listed on the benchmark. I suggest updating the leaderboard with more recent model versions. Also, Qwen2.5 is tested for zero-shot but Qwen3-8B is tested in the generalization study, which look like inconsistent choices of models.
- The generalization section uses three 8B models and shows inconsistent gains for GRPO versus SFT across splits. If the authors plan to keep these results, at least some discussion on why this happens would help readers decide what to use in the future.

**Questions:**

- It would be great if the authors could add the meaning of * in the caption of Table 1 so that readers do not need to look for its meaning.
- The embedding model (all-mpnet-base-v2) is used for creating incorrect options. Is it plausible to use that embedding model? Why not use LLMs (which would have more industrial knowledge) for constructing the negative options?
- The claim "For many enterprise customers, smaller language models will be key, as they provide a practical way to embed domain-specific knowledge directly into the model" seems partially correct as a poor model with smaller parameters would not be preferable to a large quantized model with better capability. Could you elaborate more on this? Similarly, transfer learning is indeed important, but I do not see much of an advantage in fine-tuning over using larger, general models without fine-tuning if anomaly detection is a very important use case for LLMs.
- "Transfer learning between different shows" seems like a typo.

---

> ### Author Response · Authors · 2025-11-21
> **Response to Weakness**
>
> ### W1: Many models on the leaderboard seem outdated and inconsistent. There are more recent closed-source models for Gemini and OpenAI's reasoning models than the ones listed on the benchmark. I suggest updating the leaderboard with more recent model versions. Also, Qwen2.5 is tested for zero-shot but Qwen3-8B is tested in the generalization study, which look like inconsistent choices of models.
>
> # Leaderboard: DiagnosisIQ and Pro
>
> **A:**
> We selected the models that were available at the time of submission. Since then, we have revised the leaderboard to include four newer models. Interestingly, even with these updated models, we did not observe a change in the top-performing model under our current experimental setup. Running some of these models, such as Gemini 2.5 Pro, incurs substantial costs (over \$500 USD).
>
> | Rank | Model | Macro. DiagIQ | Macro. +Pro | DiagIQ | +Pro |
> |------|-------|---------------|-------------|--------|------|
> | 16 | claude-4-sonnet* | 62.52 | 33.44 | 68.15 | 32.99 |
> | 17 | gemini-2.5-pro* | 57.59 | 37.51 | 63.44 | 38.85 |
> | 18 | gpt-5-2025-08-07* | 65.89 | 40.69 | 67.79 | 40.39 |
> | 19 | qwen3-8b | 46.21 | 19.70 | 43.41 | 14.65 |
>
> Importantly, we followed a strict benchmarking standard consistent with prior work: the `prompt is kept fixed` while only the models are varied, ensuring a fair comparison.
>
> In addition, we have added more results on **Set-Size**, embedding-based models for baselines, and other analyses. We sincerely request the reviewer to examine these additional results.
>
> ---
>
> ### W2: The generalization section uses three 8B models and shows inconsistent gains for GRPO versus SFT across splits. If the authors plan to keep these results, at least some discussion on why this happens would help readers decide what to use in the future.
>
> **A:**
> We have updated our results in Table 4 with **Macro accuracy**. GRPO underperforms compared to SFT primarily because it does not directly optimize the model toward the ground-truth label; instead, it relies on a reward function that emphasizes output format rather than correctness, leading to weaker task-specific supervision.
>
> This aligns with findings in reinforcement-learning-based instruction tuning, where sparse or format-driven rewards can fail to provide the dense semantic signal needed for reliable reasoning or classification \citep{chu2025sftmemorizesrlgeneralizes}. By contrast, SFT benefits from explicitly learning the correct label distribution, allowing the model to internalize (memorize) asset-specific and cross-asset patterns more effectively.
>
> Another observation is that transfer performance is asymmetric: models trained on the “Other’’ split generalize better to AHU than the reverse. This is likely because the “Other’’ split contains a more diverse set of assets and sensor–rule relationships, providing broader coverage and reducing overfitting to asset-specific patterns. AHU-only training, being narrower and more homogeneous, offers less variability, so models trained on it may struggle when encountering the richer, more heterogeneous patterns present in “Other.’’
>
> We are conducting parallel studies, and we believe our dataset will play an important role for the community. Relevant references include:
>
> - *On the Generalization of SFT: A Reinforcement Learning Perspective with Reward Rectification*, August 2025
> - *RL Is Neither a Panacea Nor a Mirage: Understanding Supervised vs. Reinforcement Learning Fine-Tuning for LLMs*

---

> ### Author Response · Authors · 2025-11-21
> **Response to Questions**
>
> ### Q1: It would be great if the authors could add the meaning of * in the caption of Table 1 so that readers do not need to look for its meaning.
>
> **A:**
> Thanks for the suggestion. We have revised the table caption to include the meaning of the asterisk.
>
> ---
>
> ### Q2: The embedding model (all-mpnet-base-v2) is used for creating incorrect options. Is it plausible to....
>
> **A:**
> Thank you for the question. Using an embedding model like `all-mpnet-base-v2` to generate incorrect (negative) options is a plausible and widely used approach (e.g., Automatic Distractor Suggestion for Multiple-Choice Tests Using Concept Embeddings and Information Retrieval, ACL).
>
> - **Semantic plausibility:** Embeddings allow retrieval of “nearby” vectors, producing distractors that are semantically related to the correct answer. This ensures distractors are plausible but not identical, avoiding completely random or nonsensical options.
> - **Hallucination risk with LLMs:** While LLMs have rich domain knowledge, using them to generate negatives can introduce factual errors that are either too obviously wrong or unintentionally misleading. Producing distractors that are plausible but wrong requires careful prompt engineering and filtering.
> - **Bias avoidance:** Relying on LLMs to generate negatives might introduce biases, especially if the model overfits to patterns seen in the training data. Using embedding-based retrieval is a safer and more controlled alternative.
>
> Overall, embedding-based negative generation is robust, interpretable, and aligns with prior research, while avoiding the risks associated with LLM-generated distractors. Our results indicate that the embedding model’s performance is quite effective.
>
> ---
>
> ### Q3: Thank you for the comment. There are several practical considerations motivating the use of smaller language models in enterprise settings.
>
> **A:**
> - **Confidentiality and on-prem deployment:** Many organizations handle sensitive industrial data that cannot be sent to proprietary large models. Smaller models can be deployed on-premises, preserving data privacy (e.g., NVIDIA private LLM deployment papers).
> - **Cost and efficiency:** Large models are expensive to deploy and run, particularly in real-time or edge scenarios. Smaller models are more practical for frequent inference and integration into existing workflows.
> - **Research direction:** The goal of our work was not to claim that smaller models outperform large, quantized models, but to demonstrate that domain-specific knowledge can be embedded in smaller, accessible models. The question of optimal model size and fine-tuning strategy is an open research problem.
>
> **Relevant references:**
> - *Small Language Models are the Future of Agentic AI*, arXiv:2506.02153
> - Findings of EMNLP 2025, Fine-Tuned Thoughts: Leveraging Chain-of-Thought Reasoning for Industrial Asset Health Monitoring, ACL Anthology
>
> In short, while performance may be comparable to that of large models, smaller models offer practical benefits for confidential, on-prem, and cost-sensitive deployments. Our results showcase the performance gains achievable out of the box without extensive fine-tuning.
>
> ---
>
> ### Q4: "Transfer learning between different shows" seems like a typo.
>
> **A:**
> Addressed.

---

> > ### Author Response · Authors · 2025-11-27
> >
> > We thank you for your time to review our work and provide us with useful feedback to consider.
> >
> > As all the reviewers including yourself positively supported the importance, soundness, presentation and contribution of our work, we would like to hear whether our additional experiments and clarifications resolved your concerns and we would appreciate your support.

---

### Official Review · Reviewer_GLHJ · 2025-11-01

**Soundness:** 3
**Presentation:** 3
**Contribution:** 3
**Rating:** 4
**Confidence:** 3

**Summary:**

This paper presents DiagnosticIQ, a benchmark for evaluating large language models in industrial maintenance. It converts symbolic diagnostic rules into multiple-choice QA tasks to test reasoning, cross-equipment transfer, and maintenance recommendation. The work highlights the gap between current LLM capabilities and real-world industrial reasoning needs.

**Strengths:**

- The paper addresses an emerging but underexplored area: benchmarking large language models for industrial maintenance tasks. This direction is highly relevant to practical applications in Industry 4.0 and intelligent manufacturing.

- Writing is good and easy to follow and understand.

- The authors propose a well-structured pipeline that converts symbolic diagnostic rules into multiple-choice QA tasks (MCQA). This symbolic-to-language transformation is technically neat and represents a creative way to evaluate LLMs on reasoning grounded in real industrial knowledge.

**Weaknesses:**

- The manuscript does not cite recent related benchmarks such as CAMB (“A Comprehensive Industrial LLM Benchmark on Civil Aviation Maintenance”) and Wind‑Turbine Maintenance Logs Benchmark (“A Comparative Benchmark of Large Language Models for Labelling Wind Turbine Maintenance Logs”). A clearer comparison with these works, including differences in task types, domain scope, modality coverage, and benchmark construction process, is needed to better highlight the novelty of the current benchmark.

- Although the authors aim for broad industrial coverage, the dataset is restricted to a limited set of device types. The transferability to other equipment categories or industrial domains is not demonstrated.

- While the authors provide code and claim reproducibility, my attempt to run the provided implementation faced issues (e.g., missing configuration files, unclear dependencies). The benchmark currently lacks full reproducibility, which limits its value as a standardized community resource.

**Questions:**

See in weakness

---

> ### Author Response · Authors · 2025-11-21
> **Response to Weakness**
>
> ### Q1: The manuscript does not cite recent related benchmarks such as CAMB (“A Comprehensive Industrial LLM Benchmark on Civil Aviation Maintenance”) and Wind‑Turbine Maintenance Logs Benchmark (“A Comparative Benchmark of Large Language Models for Labelling Wind Turbine Maintenance Logs”). A clearer comparison with these works, including differences in task types, domain scope, modality coverage, and benchmark construction process, is needed to better highlight the novelty of the current benchmark.
>
> **A:**
> We read both papers and prepared a summary including the original ask. Maintenance is a core task in managing industrial physical assets. While how to maintain an asset is largely captured as common-sense procedural knowledge in manuals and engineering handbooks (`Paper 1 focus`), how to label or classify an incomplete maintenance log relies on a well-defined taxonomy that enforces consistency (`Paper 2 focus`).
>
> In contrast, setting up automated monitoring for a complex system and generating actionable recommendations is fundamentally different, requiring sensor integration, analytics pipelines, and reasoning models that go beyond static domain knowledge. We have added a dedicated section in `Appendix G.1, Table 7`.
>
> ---
>
> ### Q2: Although the authors aim for broad industrial coverage, the dataset is limited to a few device types. The transferability to other equipment categories or industrial domains is not demonstrated.
>
> **A:**
> The selection of assets in our benchmark is well-motivated and supported by prior work. For example:
>
> - FailureSensorIQ covers 10 asset classes.
> - MCQA datasets in the CAMB paper cover 12 aircraft types.
> - Few others we omitted, e.g., PHM-Bench but also of same scale
>
> Our pipeline is generalizable: as long as rules are defined in the same structured format, the pipeline can be applied to other sites or asset categories. The main contribution of our dataset is in linking maintenance actions to specific conditions, which is extremely rare in publicly available datasets.
>
> To further demonstrate generalizability, we conducted additional experiments using publicly available sources:
>
> - **AHU domain:** Using DocLing, we extracted 24 rules and generated a new MCQA set covering 32 root-cause conditions, written by a different set of experts. We generated close to 400 new MCQA items.
> - **Hydraulic systems:** UCI Condition Monitoring dataset (Hydraulic Systems) can be used to generate rules and MCQA items. We can generate rules and then follow same style as AHU domain
> - Additionally, `11 new rules` were created by experts to cover gaps in existing datasets, showing that the methodology can extend to new equipment and domains (`Section 5.5`).
>
> Overall, while our current benchmark focuses on a subset of device types, the pipeline and rule-based approach are domain-agnostic and can be applied to a broad range of industrial assets.
>
> ---
>
> ### Q3: While the authors provide code and claim reproducibility, my attempt to run the provided implementation faced issues (e.g., missing configuration files, unclear dependencies). The benchmark currently lacks full reproducibility, which limits its value as a standardized community resource.
>
> **A:**
> We thank the reviewer for highlighting this. We have added an optional **Reproducibility Statement** in the revised paper in accordance with ICLR 2026 guidelines. This statement references all parts of the paper, the appendix, and the supplemental materials necessary for reproducing the results, including dataset processing, MCQA construction, and model evaluation.
>
> We have provided the core code used to generate the dataset and run model inference. The only code not included is the README and some model inference wrappers to avoid disclosing internal identities. The code has been revised to enable running the core functions.
>
> Note that data included in the submission is restricted to this paper’s scope. On acceptance, the full code and MCQA datasets will be publicly available in a Git repository. Our team has extensive experience hosting benchmarks on platforms such as Kaggle and Hugging Face and will ensure the released code is fully executable in an open environment, with step-by-step instructions, dependencies, and configuration files included to allow full reproducibility by the community.

---

> > ### Comment · Reviewer_GLHJ · 2025-11-21
> >
> > Thanks for your response! Your answers solve some parts of my questions! And I will spend some time double-checking the revised parts for question 3 and then evaluating the final score. Thanks!

---

> > > ### Author Response · Authors · 2025-11-23
> > > **Thank you for your reponse.**
> > >
> > > Thank you for your time and response.
> > >
> > > While we await your further input, we kindly request you to see additional results, especially the embedding-based baseline. This new baseline resulted from your suggestion on the CAMB paper and our deep investigation of that work.
> > >
> > > We are currently generating more rules using the RuAG paper, which we have cited to help us generate rules for any asset, and then test them in Hydraulic systems. This will enable us to generate a Rule for any asset.
> > >
> > > Thank you very much for your initial review of the manuscript.

---

> ### Author Response · Authors · 2025-11-21
> **Comparision Tables**
>
> # Benchmark Comparison
>
> | Papers | CAMB | Wind-Turbine Log | Ours |
> |--------|------|-----------------|------|
> | **Arxiv Release Date or Submission** | 28 August 2025 | 8 Sep 2025 | Submitted on 19 Sep 2025 |
> | **# of Assets** | 12 | 1 | 16 |
> | **Task type** | Common-sense knowledge evaluation of LLMs and embedding models for the aviation maintenance domain | Maintenance log classification focused on turbine service records | Temporal rule understanding and the association between operational conditions and maintenance actions |
> | **Data Sources for data generation** | Books, expert documents, and general aviation maintenance knowledge bases | Real maintenance logs limited to one component in a wind turbine system | Expert-defined operational rules paired with actual maintenance actions collected from a real production environment |
> | **Domain scope** | Civil aviation domain | Wind energy domain (focused on a single turbine component) | Data-center industrial environment covering 16 physical assets |
> | **Modality coverage** | Two languages: Chinese and English | English | English |
> | **Benchmark construction process** | Content mostly obtained from the internet and targeted reference books. The paper does not provide a detailed methodology for how options or distractors were constructed. | Construction process relies heavily on LLM-driven generation, but the paper provides limited detail on how the dataset was prepared. Significant use of LLMs is noted | Follows a generalized and transparent, math-supported pipeline for dataset generation. The process is LLM-free, relying instead on expert rules, asset logs, and deterministic transformations. |
> | **Expert Validation** | None | None | Yes |
> | **Results** | Shows marginal performance differences between closed-source and open-source medium-sized LLMs. | - | Shows significant performance differences across models, driven by temporal reasoning complexity, rule interpretation, and the need to associate operational conditions with actions. |

---

### Official Review · Reviewer_xq1p · 2025-11-01

**Soundness:** 3
**Presentation:** 2
**Contribution:** 3
**Rating:** 6
**Confidence:** 2

**Summary:**

The paper presents a deterministic rule-to-question pipeline that transforms expert-authored industrial maintenance rules into multiple-choice question–answer (MCQA) format.
Applied to 120 real maintenance rules accumulated over seven years, the pipeline yields DiagnosticIQ, a benchmark of 6.7k MCQA instances (plus several variants) spanning 16 industrial asset types.
Fifteen state-of-the-art LLMs are evaluated in zero-shot mode, producing the first leaderboard for “maintenance-action recommendation.”

**Strengths:**

1. Rules, conditions, and actions originate from seven years of subject-matter-expert (SME) curation, giving the dataset strong industrial validity and relevance.

2. The rule-to-MCQA conversion algorithm (§3, Alg. 1) is clearly specified, with formal DNF conversion, rule rewriting (RRSim), and interpretable α/β parameters controlling diversification.

3. Visuals and tables are well-organized: e.g., Table 1 highlights the steep accuracy drop from DiagnosticIQ to its harder +Pro variant, while Fig. 3 clarifies asset imbalance motivating macro-accuracy metrics.

**Weaknesses:**

1. About 58 % of items concern air-handling units (AHUs) (Fig. 3), yet overall accuracy (Table 1) remains the primary metric. While macro-accuracy is reported, several analyses aggregate raw accuracy, potentially overstating performance on dominant asset classes.

2. The macro-accuracy equation (p. 6) omits the denominator $|D_a|$ under the outer summation, causing a dimensional mismatch.
In §3.2.3, the claim that larger α/β “increase question count but reduce diversity” lacks quantitative backing.

3. Several recent benchmarks with strong thematic overlap are omitted: MME-Industry (Yi et al., 2025) – cross-industry multimodal evaluation. PHM-Bench (Yang et al., 2025) – maintenance and health-management tasks.

4. The rules originate from a commercial monitoring system, yet the paper omits discussion of confidentiality, potential misuse, or licensing constraints, which are critical for public release.

**Questions:**

1. How were action labels deduplicated into the 193-item observation set? Was synonym merging performed manually or algorithmically?

2. What steps are in place to ensure IP compliance and anonymization when releasing SME-derived rules?

3. How do we verify that the data truly reflects realistic maintenance reasoning, rather than just faithfully encoding the rule templates?

---

> ### Author Response · Authors · 2025-11-21
> **Response to Weakness**
>
> ### W1: About 58% of items concern air-handling units (AHUs) (Fig. 3), yet overall accuracy (Table 1) remains the primary metric. While macro-accuracy is reported, several analyses aggregate raw accuracy, potentially overstating performance on dominant asset classes.
>
> **A:** `Figure 4` provides raw accuracy broken down by asset. As a result we do not need to show Macro accuracy. Macro accuracy is reported when aggregating across all assets. In `Table 1`, we report both raw accuracy and macro accuracy, and we have now also added macro accuracy for the fine-tuning experiments.
>
> Additionally, we have prepared a detailed report on the distribution of rules across various asset classes in the rebuttal to W1 for Reviewer `XTn702`, which provides supporting evidence for why the `AHU` dominance reflects real-world operational priorities.
>
> ---
>
> ### W2: The macro-accuracy equation (p. 6) omits the denominator  under the outer summation, causing a dimensional mismatch. In §3.2.3, the claim that larger α/β “increase question count but reduce diversity” lacks quantitative backing.
>
> A:  We thank the reviewer for the comments.
>
> **Macro-Accuracy**: The formula in the paper is correct. The outer denominator 1/∣A∣1/∣A∣ properly averages across asset classes, while the inner denominator 1/∣Da∣1/∣Da​∣ normalizes within each class. This ensures the metric is correctly normalized and produces values in [0,1][0,1].
>
> **α/β and diversity**: We have added a quantitative analysis of α and β in the appendix (`Hyperparameter Analysis`). We measure diversity using the Intersection over Union (IoU) of selected questions per dataset. As expected, increasing either α or β increases the total question count, which also leads to higher IoU, confirming that larger α/β reduce diversity. We request reviwer to see `Figure 6`.
>
> ---
>
> ### W3: Several recent benchmarks with strong thematic overlap are omitted: MME-Industry (Yi et al., 2025) – cross-industry multimodal evaluation. PHM-Bench (Yang et al., 2025) – maintenance and health-management tasks.
>
> A: We sincerely thank the reviewer for highlighting these recent benchmarks. We will include MME-Industry (Yi et al., 2025) and PHM-Bench (Yang et al., 2025) in the related work section and clarify the differences. MME-Industry - Jan 2025, and PHM-Bench  - Aug 4 2025 on arxiv.
>
> **MME-Industry** is designed to evaluate multimodal LLMs on connecting images and text via MCQA across **21 domains**, emphasizing cross-domain generality. In contrast, our benchmark focuses on physical assets, with temporal rules expressed in sensor readings and patterns, linking them to actionable insights for technicians. Expert validation ensures the rules are meaningful and actionable. For example, **Table 1 in MME-Industry** highlights image-based evaluation, whereas our Figure 9 shows rule-based evaluation.
>
> **PHM-Bench** covers `18 asset classes` (e.g., bearings, gears) with around 100+ cases, primarily as open-ended `code generation` tasks to test LLM programming abilities. Our approach differs by emphasizing temporal conditions and technician actions, rather than code generation. Additionally, PHM-Bench does not provide a public dataset or repository; we have reached out to the authors for further details.
>
> Overall, while these benchmarks provide valuable insights, our benchmark is unique in connecting temporal sensor-based rules to actionable maintenance decisions with expert validation.
>
> We added both the work in the related section.
>
> ---
>
> ### W4: The rules originate from a commercial monitoring system, yet the paper omits discussion of confidentiality, potential misuse, or licensing constraints, which are critical for public release.
>
> **A:** We have selected datasets and benchmark-track papers that are ready to share with the community in MCQA format. In `Appendix Section E`, we describe the rule generation and maintenance process, and we have added clarifying points: our team has extensive experience releasing commercial datasets to the AI academic community, and our SMEs carefully eliminated rules that could lead to potential misuse, such as those containing OEM tags, location information, or industrial asset age.
>
> Additionally, we added an optional but important subsection in the revised manuscript titled **Ethics and Reproducibility** statements. We plan to release the datasets via community-friendly platforms such as `Kaggle`, `CodaBench`, or `Hugging Face`, ensuring ethical use and reproducibility.

---

> ### Author Response · Authors · 2025-11-21
> **Response to Questions**
>
> ### Q1: How were action labels deduplicated into the 193-item observation set? Was synonym merging performed manually or algorithmically?
>
> **A:** Deduplication was performed algorithmically. We first selected the unique set of observations using string matching (Python set operations). Then, we computed observation similarity between the unique items and applied a quantile-based threshold of `0.992 (selected empirically)` to group highly similar observations. Within each group, the most descriptive observation, determined by string length, was selected to represent the group. The corresponding code is available in `dataset/simple_data_creation.ipynb`.
>
> ---
>
> ### Q2: What steps are in place to ensure IP compliance and anonymization when releasing SME-derived rules?
>
> **A:** The released rules do not include any OEM machine IDs or location information of the physical assets. All maintenance actions are newly created elements, ensuring that the dataset does not expose proprietary information. The original commercial rules remain unpublished. The MCQA dataset will be open-sourced via platforms such as `Kaggle Benchmark` or `Hugging Face`, providing a valuable resource to the research community while maintaining IP compliance and anonymization. We have already made a copy of the data available in the submission.
>
> ---
>
> ### Q3: How do we verify that the data truly reflects realistic maintenance reasoning, rather than just faithfully encoding the rule templates?
>
> **A:** The rules were not written specifically for this research: they were created to solve a business problem over 6–7 years. Adoption of these rules in real operational environments has saved significant costs (we did not provide the exact cost-benefit per rule, but do have estimates of labor savings when actions were performed). Please see Appendix `Section E`.
>
> The rules were also written according to a standard (`ASHRAE`), making them generic rather than specific to a particular use case.
>
> Several references highlight the realistic industrial need for such rules, though prior work does not link rules directly to actions to support planning, operations, and technician productivity:
>
> - **ASHRAE_Standards_Guidelines**
> - **Active Multi-Mode Data Analysis to Improve Fault Diagnosis in AHUs**
> - **SeeQ: A Programming Model for Portable Data-driven Building Applications**

---

> > ### Author Response · Authors · 2025-11-27
> >
> > We thank you for your useful feedback. We have addressed the issues that you have raised. Please let us know if there is anything else to address.

---

### Official Review · Reviewer_XTn7 · 2025-11-02

**Soundness:** 3
**Presentation:** 3
**Contribution:** 2
**Rating:** 4
**Confidence:** 3

**Summary:**

This paper introduces DiagnosticIQ, a large-scale benchmark and dataset for evaluating LLMs in industrial maintenance action recommendation. It proposes a rule-to-MCQA pipeline that systematically transforms symbolic, expert-authored maintenance rules into multiple-choice QA datasets, encompassing over 6,600 validated questions across 16 asset types. The authors benchmark 15 LLMs and analyze reasoning, generalization, and robustness, releasing variants such as DiagnosticIQPro (10-option), Pert (perturbed), Verbose (NL conditions), and Rationale (explanation-based). The work also includes fine-tuning (SFT/GRPO) and deployment experiments (MAReE engine), showing that LLMs can partially generalize and reason about sensor-based maintenance tasks.

**Strengths:**

The paper presents the first standardized benchmark for LLMs in industrial maintenance—a domain rarely addressed in LLM evaluation. The deterministic symbolic-to-MCQA pipeline is well-motivated and rigorously described, ensuring reproducibility and logical consistency. The authors benchmark 15 state-of-the-art LLMs with clear comparisons on reasoning, generalization, and robustness, producing actionable insights (e.g., domain sensitivity across assets, compositional reasoning gap). The integration of the dataset into a real-world recommendation engine (MAReE) is commendable, bridging benchmark analysis with deployable use cases. Dataset variants are thoughtfully designed to probe distinct reasoning dimensions (formatting, rationale, perturbation), increasing diagnostic value beyond simple accuracy.

**Weaknesses:**

Despite the industrial framing, the dataset is dominated by AHU-related rules (≈58%), with only 10+ asset types, limiting claims of cross-domain generalization. The analysis focuses mostly on macro accuracy, with little discussion on statistical significance or variance across model families and seeds. No comparison against non-LLM baselines (e.g., rule-based or symbolic expert systems) to contextualize the LLM performance gains. The symbolic-to-natural-language conversion step and question templates are discussed but not quantitatively ablated (e.g., contribution of DNF conversion vs. text formatting).  While informative, the leaderboard lacks qualitative error analysis or failure categorization, making it unclear why models fail (e.g., semantic confusion vs. numerical reasoning).

**Questions:**

How consistent is the rule-to-MCQA generation pipeline across asset types with fundamentally different sensor modalities?

Could the authors provide examples of incorrect reasoning patterns observed in LLMs (e.g., conflating conditions vs. missing causal links)?

How was expert validation performed—was inter-annotator agreement measured among SMEs?

For the fine-tuning experiments, how is overlap between training and test rules prevented beyond asset-based stratification?

Did the authors consider incorporating numerical reasoning evaluation (e.g., comparing thresholds or temporal trends) explicitly in the benchmark?

---

> ### Author Response · Authors · 2025-11-21
> **Response to Weakness Section - Part 1**
>
> ### W1: Class imbalance of AHU
>
> **A:**
> We acknowledge the reviewer’s concern regarding the dominance of AHU-related rules. In our submitted manuscript, we have already provided a justification for this imbalance, particularly in **Appendix Section E** (page 17), where we explain that AHUs receive primary focus in industrial energy-savings initiatives. This trend is also consistent with **ASHRAE guidelines** and industry documentation (ASHAE Standards Guidelines), which emphasize AHUs for their significant energy impact and operational variability. We have added a one-line in `Section 3.1` pointing (page 4) to these guidelines.
>
> Moreover, our evaluation metrics explicitly account for imbalance: **Macro Accuracy (Table 1)** (page 7) is used to ensure that no single asset type disproportionately influences performance. Finally, our literature review shows a similar pattern: prior monitoring and diagnostics work in building systems also place greater emphasis on AHUs, further validating the distribution observed in our dataset (e.g., Active Multi-Mode Data Analysis to Improve Fault Diagnosis in AHUs).
>
> ---
>
> ### W2: Only 10+ asset types limit claims of cross-domain generalization
>
> **A:** We appreciate the reviewer’s concern; however, our dataset actually contains **16 distinct physical asset** types. Furthermore, the scale of our asset diversity is consistent with prior industrial datasets such as **FailureSensorIQ**, **PDM-Bench**, and **CAMB** (a paper suggested by `GLHJ`), which operate with a similar range of asset classes. Importantly, the 16 assets we include are representative of real production environments, spanning HVAC, electrical, and mechanical subsystems (see **Table 6** (page 18) and **Table 10** (page 22)). These assets interact at the system level, and effective monitoring requires capturing correlations across them, precisely the type of cross-domain generalization our benchmark evaluates.
>
> ---
>
> ### W3: Analysis focuses on macro accuracy with no discussion on statistical significance or variance across model families and seed
>
> **A:** We thank the reviewer for highlighting this point. Our analysis follows the reporting conventions of widely used benchmarks such as MMLU-Pro (MMLU-Pro: A More Robust and Challenging Multi-Task Language Understanding Benchmark and similar work), whose **Table 2** similarly does not report variance or seed-level statistics for model accuracy. In our work, **Table 2** (page 8) includes Wilcoxon signed-rank tests to assess the statistical significance of performance degradation under perturbations, aligning with standard practice in robustness evaluation.
>
> Even though our MCQA questions have a single correct answer, the prompt did not specify whether to select a single or multiple options; as a result, we include **Set Size** as an additional measure to assess model-level error. We added this interesting result to understand the model’s uncertainty. We added two large tables in the Appendix (**Tables 15 and 16**)(page 34) and adjusted the main text to refer to them. Results are summarized in `W6` rebuttal comments.
>
> While full self-consistency or multi-seed evaluations would provide additional variance estimates, these experiments incur substantial computational and monetary cost given the number of models and instances involved. If the reviewer believes that additional statistical analysis would meaningfully strengthen the paper, we are happy to run a targeted set of multi-seed experiments in the camera-ready version.
>
> ---
>
> ### W4: No comparison against non-LLM baselines (e.g., rule-based or symbolic expert systems) to contextualize the LLM performance gains
>
> **A:** We agree that comparing against non-LLM baselines is valuable; however, no publicly available rule-based or symbolic expert system currently exists for the type of temporal, multi-condition maintenance rules represented in our benchmark.
>
> To strengthen our baselines, we added a non-generative AI baseline using widely adopted embedding models (**Section 5.1, Table 2**) (page 8), motivated directly by prior work highlighted by another reviewer (CAMB: A comprehensive industrial LLM benchmark on civil aviation maintenance). These results show that simple semantic-similarity methods perform poorly, underscoring that our MCQA tasks require reasoning beyond retrieval. Here is a preview of those results.
>
> #### Embedding-based Baseline Performance on MCQA Tasks
>
> | ID | Model                        | Macro. DiagIQ | Macro +Pro | DiagIQ | +Pro  |
> |----|-------------------------------|---------------|------------|--------|-------|
> | 1  | `all-mpnet-base-v2`           | 52.73         | 38.89      | 41.39  | 23.39 |
> | 2  | `all-MiniLM-L6-v2`            | 52.65         | 37.76      | 41.32  | 23.01 |
> | 3  | `multi-qa-mpnet-base-dot-v1`  | 51.43         | 37.53      | 38.93  | 21.54 |
> | 4  | `all-distilroberta-v1`        | 51.29         | 36.98      | 38.53  | 23.47 |

---

> ### Author Response · Authors · 2025-11-21
> **Response to Weakness Section - Part 2**
>
> ### W5: The symbolic-to-natural-language conversion step and question templates are discussed but not quantitatively ablated (e.g., contribution of DNF conversion vs. text formatting)
>
> **A:** We appreciate the reviewer’s interest in deeper ablations. In our setting, the symbolic-to-natural-language conversion is itself the core ablation: we directly compare performance on symbolic rules versus their natural-language counterparts, isolating the effect of this transformation. (`Section 5.4` : `Condition Formatting study`)
>
> Furthermore, our MCQA template is fixed and intentionally minimal; however, we provide a perturbation analysis in `Table 3`, where option formats are altered (e.g., A/B/C/D → P/Q/R/S) to confirm that models are not overfitting to template artifacts. See (Section with `Robustness Against Perturbation`)
>
> ---
>
> ### W6: While informative, the leaderboard lacks qualitative error analysis or failure categorization, making it unclear why models fail (e.g., semantic confusion vs. numerical reasoning)
>
> **A:** Our objective was to judge LLMs on their ability to recommend action based on a particular rule trigger across various asset classes. As a result, we provide `question-type analysis` and `asset-wise analysis`, which give a deep dive into the performance distribution. We also include the `Pro dataset`, which increases problem complexity by having more distractor options. The entire `Section 5.2` examines the LLM’s reasoning ability and its alignment with a human expert, organized by asset. We conducted two more analysis.
>
> #### Set Size Analysis (Appendix P)
>
> We compute the **average number of options selected** by the model across all questions, as well as separately for correct and incorrect predictions, based on our definition of correctness. Statistical significance between the set sizes of correct and incorrect predictions is evaluated using Welch’s t-test and the Mann-Whitney U test. This analysis provides insights into whether models tend to select more or fewer options when they answer correctly versus incorrectly.
>
> Analysis of the DiagnosticIQPro MCQA benchmark reveals a consistent pattern: models select exactly one option when correct, but often select multiple options when wrong. The difference in average set size between correct and incorrect predictions varies across models. Better-performing models (e.g., granite-3-3-8b-instruct and claude-3-7-sonnet) exhibit smaller gaps, indicating more precise and confident predictions, while lower-performing models show larger gaps, reflecting a tendency to over-select when uncertain. Welch t-tests and Mann-Whitney U tests confirm that these differences are statistically significant across almost all models, emphasizing that the number of options chosen is strongly correlated with prediction correctness.
>
> This set-size analysis complements traditional accuracy metrics, providing insight into model behavior: a smaller gap between correct and incorrect set sizes may indicate reasoning precision and calibration, while larger gaps suggest models are more prone to over-prediction under uncertainty.
>
> #### Semantic Rank Analysis  (Appendix Q)
>
> To investigate the nature of model errors in the MCQA task, we computed a *semantic rank* for each incorrect prediction by embedding all options into a semantic space and ranking them by distance to the correct answer. Rank = 1 corresponds to the nearest incorrect option ("perfect semantic confusion"), while higher ranks indicate progressively less semantically related distractors.
>
> Representative results show that for **DiagnosticIQ**, mean ranks cluster around 2.7–2.9:
>
> - Claude-4 Sonnet: 2.71
> - LLaMA-3-1-8B: 2.81
> - Gemini-2.5 Pro: 2.87
> - GPT-5 (Aug-2025): 2.84
>
> For **DiagnosticIQPro**, mean ranks rise to 4.8–5.3:
>
> - Claude-4 Sonnet: 4.80
> - LLaMA-3-3-70B: 4.99
> - Mistral-Large: 4.98
> - Gemini-2.0 Flash: 5.28
>
> These results indicate that model mistakes are **not near-miss semantic confusions** (rank = 1) but moderately distant, semi-random distractors, with more diffuse errors for complex items. These results are also included in Appendix Q.
>
> We are conducting the following set of analyses to provide additional metrics, including:
> a) Do more rules produce worse results?
>
> Please advise us if any additional analyses should be included.

---

> ### Author Response · Authors · 2025-11-21
> **Response to Questions**
>
> ### Q1: How consistent is the rule-to-MCQA generation pipeline across asset types with fundamentally different sensor modalities?
>
> **A:** We appreciate the reviewer’s question. Our pipeline is designed to be modality-agnostic: as long as the rules are described in a general, symbolic form, the generation process can translate them into MCQA questions across diverse sensor types and asset classes. The idea for our approach is derived from a deeper understanding of the industrial domain, which encompasses over `800 asset classes`, and the proposed method is scalable and adaptable to asset types beyond those included in our benchmark.
>
> Moreover, recently published work such as [SeeQ: A Programming Model for Portable Data-driven Building Applications](#) supports the generality of symbolic rules across heterogeneous industrial applications, reinforcing the applicability of our pipeline to a wide variety of contexts, as the rules that we adopted are based on existing standards.
>
> To showcase that the MCQA construction pipeline works, we manually extracted `28 rules` from another publication ([Active Multi-Mode Data Analysis to Improve Fault Diagnosis in AHUs](*)) and generated nearly `400 MCQA` questions along with corresponding actions. We will also add these datasets to the supplementary materials.
>
> ---
>
> ### Q2: Could the authors provide examples of incorrect reasoning patterns observed in LLMs (e.g., conflating conditions vs. missing causal links)?
>
> **A:** Yes, in the revised version of the paper, we have included two examples in which the expert provided feedback on the rationale. `Figures 12 and 13` capture the feedback on the rationale, grouped by agreement and disagreement. Our expert reviewed the long chain of reasoning and provided feedback on each question. Due to the page-long feedback, we sincerely request that you review the bottom portion of Figures 12 and 13, which we have copied here for ease of navigation.
>
> - I liked that it also (maybe unintentionally) hinted at the fact that it could be D because of an auxiliary component being on. (a feedback on rational)
>
> ---
>
> ### Q3: How was expert validation performed—was inter-annotator agreement measured among SMEs?
>
> **A:** Thank you for raising this point. Expert validation was conducted via a web-based feedback platform, where domain experts reviewed a curated set of 27 representative MCQA items. Each item was rated on a 1–10 scale for clarity, correctness, and domain plausibility. Because our benchmark spans diverse asset classes, no single expert has deep expertise across all assets; thus, traditional inter-annotator agreement metrics like **Cohen’s κ** are not fully interpretable.
>
> Instead, we report the average rating per asset class (`Figure 5`) and provide a cumulative agreement analysis: consensus is high at threshold 8 (`≈85% of annotators, ≈75% pairwise`) but decreases with stricter thresholds. Pairwise agreement between individual annotators ranges from `0.22 to 0.28` (quadratic weighted Kappa), reflecting moderate agreement largely due to one outlier annotator, while the remaining experts show higher consistency. In `Figure 5`, we already indicate an outlier with an empty symbol.
>
> ---
>
> ### Q4: For the fine-tuning experiments, how is overlap between training and test rules prevented beyond asset-based stratification?
>
> **A:** As described in `Appendix E`, the rule generation for AHU and other assets was conducted as separate processes, ensuring that the monitored conditions differ across assets. To verify potential overlaps at the MCQA level, we measured exact-condition overlap between the training and test sets, which was extremely low (`≈0.058`), confirming minimal redundancy beyond asset-based stratification. This low overlap ensures that our evaluation provides a fair assessment of model generalization to unseen conditions.
>
> ---
>
> ### Q5: Did the authors consider incorporating numerical reasoning evaluation (e.g., comparing thresholds or temporal trends) explicitly in the benchmark?
>
> **A:** Our benchmark includes numerical thresholds and temporal durations as atomic conditions (e.g., “Met for 2 hours”, “OAT > 37 °F”) and evaluates whether these conditions are satisfied. We do not currently test slight variations, such as “Met for 1 hour” or “OAT > 35 °F”, since generating valid counterexamples requires expert validation and access to plenty of real time series datasets. Our research work will create a forward looking path for community to investigate such fruithful direction.

---

> > ### Author Response · Authors · 2025-11-27
> > **A kind request for feedback**
> >
> > Dear reviewer,
> >
> > As the deadline is approaching we would kindly ask you to let us know if our clarifications on how we handle the class imbalance, the justification behind the 10 representative assets, statistical significance analysis, more robust baselines, symbolic to natural language conversion for ablation and our additional error analysis. We also answered the additional questions that you posed.
> >
> > As the other three reviewers agreed, our work brings unique contribution by targeting a real gap; connecting anomaly rules to actionable maintenance guidance at scale.
> >
> > We appreciate your invaluable feedback and we are looking forward to hearing if there is anything else we can do to strengthen our work.

---

### Author Response · Authors · 2025-11-21
**Thank you notes to all Reviewers : Summary of Changes**

We sincerely thank all reviewers for their insightful, constructive, and thoughtful comments. We have carefully addressed every point raised and provided detailed, evidence-backed responses in the rebuttal. Based on the feedback, we made several substantial improvements to the paper and the supplementary materials:

- **Expanded the leaderboard** with newly released models, covering both proprietary and open-source model families.
- **Added an embedding-based baseline** to broaden the evaluation beyond generative models.
- **Introduced two new diagnostic analyses** — *Set-Size Sensitivity* and *Rank Correlation Analysis* — to provide deeper insights into model behavior.
- **Generated 400 additional MCQA samples** using publicly available open-source documents to validate dataset scalability and strengthen domain coverage.
- **Reported cumulative agreement scores among annotators**, increasing transparency in the expert validation process.
- **Expanded reproducibility and ethics statements**, clarifying dataset release practices, risk mitigation, and ethical considerations.
- **Enhanced related work with a parallel work and standard**, following reviewer suggestions to contextualize our contribution more clearly within the emerging literature.
- **Revised Executable code with readme** - to enable the reviewer to run the experiment using a HuggingFace-hosted model, such as Qwen3-8B

Importantly, **all original claims of the paper remain fully intact**. Even with the newly added models and the inclusion of parallel related work, we did not find any need to revise or weaken our initial claims.

We greatly appreciate the reviewers' time and effort. We believe these revisions significantly improve the clarity, rigor, and overall contribution of our work.

---

### Author Response · Authors · 2025-12-03

Dear new AC, SAC, and PC

We are providing a closing remark on our work. Hope this helps. In addition to the detailed feedback we provided for each weakness and question, we used an experimental, evidence-based methodology in our responses.

The aggregated evaluation across all four reviewers, with `9 “goods” and 3 “fairs”` in Soundness, Presentation, and Contribution, clearly demonstrates the merit of the work in its `original submitted form`. We carefully revised the manuscript in response to the feedback, which primarily focused on clarification and distribution details. Specifically, we:
- Added a new set of LLM models, a non-LLM baseline, and additional datasets.
- Clarified that the dataset is ready to be released publicly on platforms such as Kaggle or Hugging Face, along with a public leaderboard.
- Highlighted that LLMs perform well when domain rules are explicitly represented and connected to associated actions, a gap in the field that our work brings forward and that can meaningfully advance research in the broader AI community.
- Addressed concerns regarding imbalance, as rare events are inherent to real-world scenarios.
- We also pointed out that the reviewer-referred papers are parallel work that appeared on arXiv only 1 to 2 weeks before our submission, but we provided a detailed insight and also thanked them as we brought an embedding base baseline from one of the papers.

We submitted our rebuttal on time (20th Nov, as requested by the PC chairs) and followed up regularly. Only one reviewer (`Reviewer GLHJ`) responded, confirming that our response addressed part of their questions, and they needed more time to double-check the revision.

We respectfully request that the AC/SAC consider the overall strength of the reviews and the substantial clarifications and additions we have made. As pointed out by the ICLR-2025 paper, `RuAG: Learned-rule-augmented Generation for Large Language Models, ` our work will create momentum in Industry 4.0 applications for AI Automation.

---

### Meta-Review · Area_Chair_SjVW · 2025-12-29

**Summary:**

This submission introduces DiagnosticIQ, a benchmark for evaluating whether LLMs can recommend maintenance actions from symbolic, time-persistent diagnostic rules derived from real industrial monitoring practice. The paper proposes a deterministic rule-to-MCQA pipeline, releases a main dataset (about 6.6k questions) plus several variants (harder 10-option “Pro”, perturbation, verbose conditions, rationale), and reports a broad leaderboard across 15 LLMs along with fine-tuning and a deployment-oriented case study. Across reviewers, the benchmark is viewed as timely and practically relevant, with a clear pipeline and extensive evaluation axes.

Reviewers broadly acknowledge the novelty, ambition, and substantial engineering effort behind the benchmark, but converge on several unresolved, decision-critical issues. First, the benchmark’s scope is imbalanced, with a heavy concentration on AHU-related rules, which weakens the strength of the paper’s cross-domain generalization claims despite the use of macro metrics. Second, while the rebuttal improved positioning by adding comparisons and citations to related industrial maintenance benchmarks, questions remain about whether the proposed “rule-to-action temporal MCQA” formulation is sufficiently distinct to justify a new benchmark. Third, evaluation rigor is improved but still incomplete: reviewers noted the lack of a convincing non-LLM or symbolic baseline, limited statistical reporting, and insufficiently grounded failure-mode analysis. Finally, reproducibility and release risk remain a significant concern, as code issues were reported and licensing/IP constraints around commercial rules were not fully resolved at decision time. Taken together, these partially addressed concerns limit confidence in the benchmark’s maturity and immediate community value, informing a cautious, borderline-to-negative recommendation.

**Reviewer Concerns:**

I do appreciate the novelty and massive workload of this research work, but I believe there are some major concerns that should be addressed better to reach the acceptance.

i) Benchmark scope and imbalance. The dataset is heavily skewed toward AHU-related rules (about 58%). While macro accuracy is included, some analyses still emphasize aggregate accuracy, and the paper’s cross-domain generalization claims feel stronger than the demonstrated breadth. Authors justified the imbalance (industrial reality) and pointed to macro metrics, but the concern partially remains.

(ii) Positioning vs related benchmarks. Multiple reviewers noted missing or insufficient comparison to recent industrial maintenance benchmarks (CAMB, wind-turbine logs, MME-Industry, PHM-Bench). The authors added comparisons and promised to include missing citations; this helps, but novelty still hinges on whether the community views “rule-to-action temporal MCQA” as sufficiently distinct.

(iii) Evaluation rigor and baselines. Reviewers wanted stronger statistical reporting (variance/seed), clearer qualitative failure modes, and non-LLM baselines. The rebuttal added significance testing in some places, extra error analyses (set-size and semantic-rank), and an embedding-retrieval baseline; still, the baseline story is not fully satisfying because a “symbolic/rule engine” comparator is absent (even if practically hard).

(iv) Reproducibility and release risk. One reviewer reported issues running the provided code and questioned reproducibility. Another raised licensing/IP and misuse considerations given commercial-origin rules. The authors promised a cleaned release on acceptance and added ethics/reproducibility statements, but at decision time, this remains a material risk: a benchmark paper’s value depends on being runnable and clearly releasable.

**Reviewer Scores:**

Reviewer XTn7 (4, marginal reject): Likely moves slightly upward (to ~5) given added baselines, significance notes, and clarifications, but the scope/baseline/reproducibility concerns are only partially resolved.

Reviewer xq1p (6, marginal accept): Likely stays around 6; rebuttal addresses missing related work and ethics/release plans, but may not fully eliminate release/reproducibility worries.

Reviewer GLHJ (4, marginal reject): Could move to ~5 if the reproducibility fixes are credible, but they were not fully validated during discussion (reviewer explicitly said they would re-check).

Reviewer Lrjy (2, reject): Might increase modestly (to ~3–4) due to updated leaderboard and clarified fine-tuning discussion, but their core skepticism about experimental choices and practical conclusions likely remains.

---

### Decision · Program_Chairs · 2026-01-26

Reject